# Systematic analysis of drug-associated myocarditis reported in the World Health Organization pharmacovigilance database

Lee S. Nguyen [1,2 ✉], Leslie T. Cooper[3], Mathieu Kerneis[4], Christian Funck-Brentano [2], Johanne Silvain[4], Nicolas Brechot[5], Guillaume Hekimian[5], Enrico Ammirati[6], Badr Ben M'Barek[1], Alban Redheuil[7], Estelle Gandjbakhch[4], Kevin Bihan [2], Bénédicte Lebrun-Vignes[2,8], Stephane Ederhy[9], Charles Dolladille[10,12], Javid J. Moslehi[11,12] & Joe-Elie Salem [2,11 ✉]

While multiple pharmacological drugs have been associated with myocarditis, temporal trends and overall mortality have not been reported. Here we report the spectrum and main features of 5108 reports of drug-induced myocarditis, in a worldwide pharmacovigilance analysis, comprising more than 21 million individual-case-safety reports from 1967 to 2020. Significant association between myocarditis and a suspected drug is assessed using disproportionality analyses, which use Bayesian information component estimates. Overall, we identify 62 drugs associated with myocarditis, 41 of which are categorized into 5 main pharmacological classes: antipsychotics ($n = 3108$ reports), salicylates ($n = 340$), antineoplastic-cytotoxics ($n = 190$), antineoplastic-immunotherapies ($n = 538$), and vaccines ($n = 790$). Thirty-eight (61.3%) drugs were not previously reported associated with myocarditis. Antipsychotic was the first (1979) and most reported class ($n = 3018$). In 2019, the two most reported classes were antipsychotics (54.7%) and immunotherapies (29.5%). Time-to-onset between treatment start and myocarditis is 15 [interquartile range: 10; 23] days. Subsequent mortality is 10.3% and differs between drug classes with immunotherapies the highest, 32.5% and salicylates the lowest, 2.6%. These elements highlight the diversity of presentations of myocarditis depending on drug class, and show the emerging role of antineoplastic drugs in the field of drug-induced myocarditis.

[1] CMC Ambroise Paré, Research & Innovation (RICAP), Neuilly-sur-Seine, France. [2] Department of Pharmacology, Sorbonne University, INSERM CIC Paris-Est, AP-HP, ICAN, Regional Pharmacovigilance Centre, Pitié-Salpêtrière Hospital, F-75013 Paris, France. [3] Department of Cardiovascular Medicine, Mayo Clinic, Jacksonville, Florida, USA. [4] Sorbonne Université, ACTION Study Group, Institut de Cardiologie, Hôpital Pitié-Salpêtrière (AP-HP), INSERM UMRS 1166, Paris, France. [5] Department of Critical Care Medicine, AP-HP, Sorbonne Université, Pitié-Salpêtrière, Paris, France. [6] "De Gasperis" Cardio Center and Transplant Center, Niguarda Hospital, Milan, Italy. [7] Department of Cardiovascular Imagery, AP-HP, Sorbonne Université, Pitié-Salpêtrière, Paris, France. [8] Université Paris Est Creteil, EpiDermE, F-94010 Créteil, France. [9] Department of Cardiology, AP-HP, Sorbonne Université, Pitié-Salpêtrière, Paris, France. [10] Department of Pharmacology, CHU de Caen, Caen F-14000, France. [11] Departments of Medicine and Pharmacology, Cardio-oncology Program, Vanderbilt University Medical Center, Nashville, TN, USA. [12] These authors contributed equally: Charles Dolladille, Javid J. Moslehi. ✉email: nguyen.lee@icloud.com; joe-elie.salem@aphp.fr

Myocarditis is characterized by inflammation of the heart muscle tissue. Clinical manifestations range from minor (isolated chest pain) to life-threatening conditions (acute heart failure, cardiogenic shock, ventricular arrhythmia)[1,2]. In 2017, there were 1.375 million incident cases of myocarditis, with most cases caused by viruses[3,4]. In contrast, drug toxicity and hypersensitivity are underdiagnosed causes of myocarditis, which may be responsible for severe and complex clinical presentation, including fulminant lymphocytic myocarditis[1,5], and allergic or hypersensitivity eosinophilic myocarditis[6]. Pharmacovigilance analyses based on spontaneous report systems allow for post-marketing drug surveillance (i.e., phase IV)[7]. So far, the identified classes of drugs associated with myocarditis included immune checkpoint inhibitors[8], antipsychotics[9,10], antibiotics, and vaccines including against COVID-19[11,12].

Yet, no systematic analysis of drug-induced myocarditis has been performed, with available studies only focusing on specific drugs. Moreover, temporal analysis of drug-induced myocarditis reporting for each class of drugs has not been done.

Here, we analyzed data extracted from the World Health Organization (WHO) global database of individual case safety reports (ICSR), aiming to identify drug substances and drug classes significantly associated with myocarditis, to describe their relative prevalence over time, and finally report the clinical features of all reported drug-induced myocarditis according to drug classes.

In this work, we highlight the existence of at least five categories of drugs: antipsychotics, cytotoxic drugs, immunotherapies, vaccines, and salicylates. These classes present distinct clinical presentation, time to onset, and subsequent mortality, suggesting class effect. These elements warrant further clinical review to confirm association and causality.

## Result

**Overall analysis.** From VigiBase® inception (1967) to January 2020, 6823 ICSR of suspected drug-induced myocarditis were reported from a total of 21,185,309 ICSR in the full database, and from 47 countries.

We included 5108 ICSR, corresponding to 62 drugs that were significantly associated with myocarditis (with a disproportionality analysis information component (IC) estimate credibility interval lower bound $IC_{025}$ above 0, Fig. 1).

The median age of affected patients was 35 [interquartile range: 25; 50] years, with men representing 73.8% (3592/4866) of cases. Median time-to-onset (TTO) was 15 [10; 23] days. Overall myocarditis-ICSR resulted in death in 10.3% (524/5108) of cases. Reporting identified a single suspected drug in 72.5% (3704/5108) of cases. Reports originated from the standard of care in 4857/5108, 95.1% and 4.9% from investigational drug studies. The summary of ICSR characteristics is detailed in Table 1. Sensitivity analysis by age subgroups and after exclusion from the full database of ICSR including drugs already identified as at known risk of myocarditis in Food and Drug Administration's (FDA) labels among these 62 drugs is available in Supplementary Dataset 1, and showed similar results.

**Drugs associated with myocarditis.** The most represented drugs were clozapine (3035/5108, 59.4%), immune checkpoint inhibitors (522/5108; 10.2%), mesalazine (311/5108, 6.1%), and smallpox-vaccine (383/5108, 7.5%) (Table 2). Among 62 drugs, 41 belonged to 5 drug classes (as defined by the Anatomical Therapeutic Chemical (ATC)), which included antipsychotic (3108/5108, 60.8%), cytotoxic drugs (190/5108, 3.7%), immunotherapies (538/5108, 10.5%), vaccines (790/5108, 15.5%), and salicylates (340/5108, 6.7%) (Table 1). Twenty-one (21/62, 33.9%)

uncategorized drugs were regrouped in a miscellaneous category (243/5108, 4.8% of ICSR); including among the most represented drugs, valproic acid (45/5108, 0.9%), and minocycline (25/5108, 0.5%). The first drug-induced myocarditis to be reported was associated with chlorpromazine in June 1979. The year when $IC_{025}$ became significant, for each liable drug substance is reported in Supplementary Dataset 2. Overlaps between drug classes within these ICSR are displayed in Fig. 2 and Supplementary Fig. 1. The number of drugs for which myocarditis signal was not previously flagged in the FDA's label was 38/62, 61.3% (including balsalazide, influenza, and hepatitis vaccines). Among these 38 drugs, 20 (52.6%) shared similar pharmacological properties as others included in drug classes known as purveyors of myocarditis, which we described in the Discussion part. We performed a separate pharmacovigilance analysis regarding the 18 incriminated uncategorized drug substances, to assess the plausibility of myocarditis due to these drugs in Supplementary Dataset 3[13].

**Cumulative cases reported.** Cumulative cases reported over time are presented in Fig. 3A. Myocarditis reports per year increased for antipsychotics from 10 in 1995 to 463 in 2019, at which point it was overwhelmingly represented by clozapine in 98.7% (457/463) of cases. More recently, immunotherapy-associated myocarditis reports increased from 6 in 2015 to 250 in 2019. Anti-PD1 drugs (cemiplimab, nivolumab, and pembrolizumab) were felt to be the culprit in 84.4% (211/250) of cases, although anti-CTLA4 inhibitors were given concomitantly in 88/211, 41.7% cases. There was a sharp increase of vaccines-associated myocarditis with 294 reports in 2010 (including 177/294, 60.2% associated with smallpox, 72/294, 24.5% with anthrax, and 48/294, 16.3% with influenza) versus fewer than 10 reports/year 5 years prior (see the evolution through time in Fig. 3B). The evolution of each drug class' proportion over decades is presented in Fig. 3C. Antipsychotics remain the most reported drug class throughout time periods, although immunotherapy has emerged as the second most reported drug class in the 2015–2020 period.

**Clinical features.** Comparisons between the five drug classes, regarding demographics and clinical features are presented in Table 2. Myocarditis associated with vaccines had younger patients versus those associated with antineoplastic drugs (cytotoxic and immunotherapy) (24 [20; 34] vs 65 [49; 73] years, $p < 0.0001$). Sex-ratio differed between drug classes: from 52.7% of cases represented by women in cytotoxic agents to 16.2% in vaccines. Drug classes associated with myocarditis were more often co-aggregating to certain specific categories of concurrent adverse drug reaction (ADR) (Table 2). Antipsychotics were more associated with hyperthermia (528/3108, 17.0%) and eosinophilia (189/3108, 6.1%) versus the 4 other class of drug-associated myocarditis (210/1854, 11.3%, $p < 0.0001$ and 19/1854, 1.0%, $p < 0.0001$, respectively). Immunotherapy drug class was less associated with pericardial manifestation (11/538, 2.0%) but more associated with myositis (128/538, 23.8%) and myasthenia-gravis (57/538, 10.6%) compared to the four other drug classes (424/4424, 9.6%, $p < 0.0001$; 362/4424, 8.2%, $p < 0.0001$, 4/4424, 0.1%, $p < 0.0001$; respectively). Cytotoxic chemotherapies were more associated with heart failure (82/190, 43.2%) and kidney disorders (16/190, 8.4%) than the four other drug classes (911/4772, 19.1%, $p < 0.0001$; 136/4772, 2.8%, $p < 0.0001$, respectively). Vaccines were more associated with pericardial reactions (154/790, 19.5%) than the four other drug classes (285/4172, 6.8%; $p < 0.0001$). More details regarding concurrent ADR by specific drug substances is available in Supplementary Dataset 4.

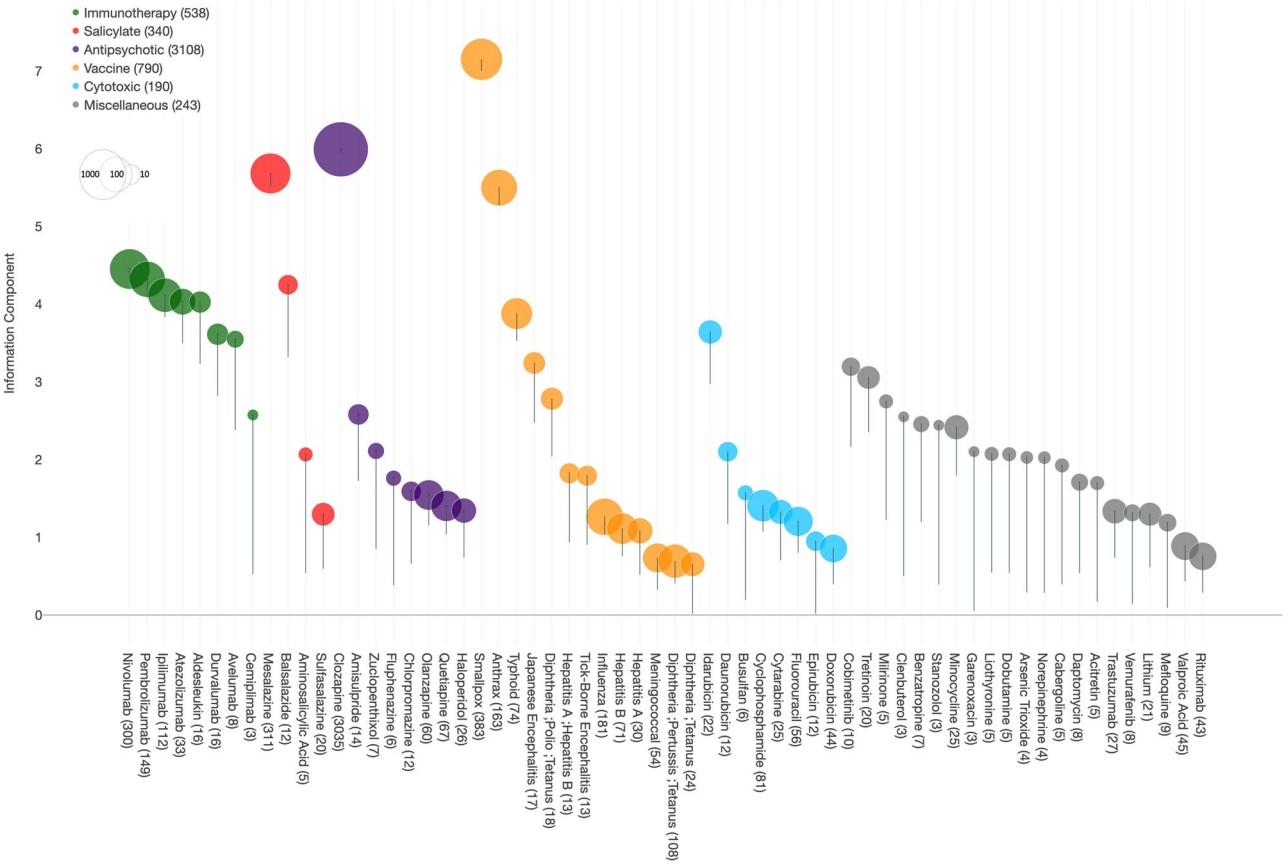

**Fig. 1 Drugs associated with myocarditis in VigiBase(r), from 1967 to Janunary 2020.** Information component and its 95% lower-end credibility interval (IC$_{025}$) (represented by the error bar) for each drug significantly associated with myocarditis (IC$_{025}$ > 0). Related raw data are available in the Source Data file.

TTO depended on the drug class (Table 2 and Fig. 3D). Vaccine-associated myocarditis had the shorter TTO, with 10 [6; 12] days, while immunotherapy-myocarditis had the longest with 33 [20; 88] days (intergroup comparison $p < 0.0001$). Cytotoxic-myocarditis had a TTO of 15 [4; 61] days, salicylate-myocarditis of 17 [12; 28] days, and antipsychotic-myocarditis of 16 [12; 22] days. Intragroup (i.e, within antipsychotics) analyses found that TTO was shorter when several antipsychotic drugs were combined, compared to one antipsychotic drug (14 [6; 21] vs. 17 [13; 23] days, $p < 0.0001$). In the other drug classes, there were no significant differences in TTO when comparing monotherapy versus combination therapy, inside each drug class (immunotherapy, $p = 0.83$, vaccines, $p = 0.90$, cytotoxic, $p = 0.36$, and salicylate, $p = 0.44$). TTO was not different between reports of lethal myocarditis versus non-lethal ($p = 0.99$).

**Mortality.** Patients who died were older with a median age of 47 [27; 64] versus 35 [25; 48] years ($p < 0.0001$). Mortality was higher in women than in men (190/1274, 14.9% vs 280/3592, 7.8%, $p < 0.0001$). Cases in which a single suspected drug was reported were less associated with death than cases in which multiple drugs were suspected involved (304/3704, 8.21% vs 220/1404, 15.7% cases, $p < 0.0001$). Immunotherapies had the highest mortality (175/538, 32.5%) and salicylates the lowest (9/340, 2.6%, $p < 0.0001$) (see Fig. 4A). Cytotoxics had a mortality of 46/190, 24.2%, antipsychotics, 181/3108, 5.8%, and vaccines, 83/790, 10.5% (see Supplementary Dataset 2 for details per drug substances). In immunotherapies, mortality was higher when two or more immunotherapy drugs (in 98/99, 99.0% of combinations were an association of anti-CTLA4 with anti-PD1) were

associated than one (46/99, 46.5% vs 129/439, 29.4%, $p = 0.001$). Mortality increase in combination therapy was also observed in the antipsychotic drug class (14/93, 15.1% vs 167/3015, 5.5%, $p < 0.001$) (clozapine was involved in most combinations with other antipsychotics, in 87/93, 93.5%). In the other drug classes, combination within each drug class was not associated with higher mortality (salicylates, $p = 1.0$; cytotoxics, $p = 0.25$). In myocarditis associated with vaccines used in monotherapy, smallpox had a mortality of 2/200, 1.0%, hepatitis A and/or B, of 12/67, 17.9%, influenza of 25/131, 19.1%, and diphtheria, tetanus, poliomyelitis, and/or pertussis vaccine of 28/109, 25.7% (overall comparison between these proportions yielded a $p < 0.0001$).

Mortality significantly differed across quartiles of time for all drug classes except for cytotoxics (evolution of mortality by quartile of time, by drug class, are presented in Fig. 4). Between the first quartile to fourth quartile of cumulative cases, mortality decreased from 55/131, 42.0% to 36/134, 26.9% in the immunotherapy drug class ($p = 0.01$); from 43/197, 21.8% to 16/200, 8.0% in vaccines ($p = 0.0001$); and from 79/776, 10.2% to 43/787, 5.5% in antipsychotics ($p = 0.0005$). Of note, minocycline (classified miscellaneous, herein) was associated with the highest mortality, with 13/25, 52%.

## Discussion

This pharmacovigilance study is a comprehensive overview of drug-induced myocarditis with analysis for over 5000 reported cases, most of which include "real-world" experience. The three most frequently culprit drug classes were antipsychotics, anticancer drugs, and vaccines, whereas clozapine, smallpox/anthrax vaccines, and nivolumab were the individual drugs associated

**Table 1 Overall characteristics of reports of myocarditis in the WHO pharmacovigilance database, VigiBase®.**

| | Number (%) | $N_{available}$ (%) |
|---|---|---|
| *Region reporting* | | All |
| APAC | 1826/5108 (35.7%) | |
| Europe | 1658/5108 (32.5%) | |
| LATAM | 9/5108 (0.2%) | |
| MENA | 22/5108 (0.4%) | |
| North America | 1592/5108 (31.2%) | |
| Sub-Saharan Africa | 1/5108 (0.0%) | |
| *Reporting year* | | All |
| 1979–1989 | 23/5108 (0.5%) | |
| 1990–1999 | 104/5108 (2.1%) | |
| 2000–2009 | 899/5108 (17.9%) | |
| 2010–2020 | 4082/5108 (81.3%) | |
| *Reporter qualification* | | 3153/5108 (61.7%) |
| Health professional | 3055/3153 (96.9%) | |
| Non-health professional | 98/3153 (3.1%) | |
| *Reporting type* | | All |
| From investigational drug studies | 251/5108 (4.9%) | |
| Non-study related (real-life) | 4857/5108 (95.1%) | |
| *Gender* | | 4866/5108 (95.3%) |
| Men | 3592/4866 (73.8%) | |
| Women | 1274/4866 (26.2%) | |
| *Age (median [IQR] (min–max) in years)* | 35 [25; 50] (0–91) | 4337/5108 (84.9%) |
| *Delay (median [IQR] (min–max) in days)[a]* | 15 [10; 24] (1–8685) | 2267/5108 (44.4%) |
| *Drug class[b]* | | All |
| Antipsychotic | 3108/5108 (60.8%) | |
| Immunotherapy | 538/5108 (10.5%) | |
| Cytotoxic | 190/5108 (3.7%) | |
| Salicylate | 340/5108 (6.7%) | |
| Vaccine | 790/5108 (15.5%) | |
| Miscellaneous | 243/5108 (4.8%) | |
| *Fatal outcomes* | 524/5108 (10.3%) | All |
| *Single drug suspected* | 3704/5108 (72.5%) | All |

*APAC* Asia-Pacific, *ICI* immune checkpoint inhibitor, *IQR* interquartile range, *LATAM* Latin America, *MENA* Middle East and North Africa.
[a]Delay is computed between first intake of treatment and first sign reported of myocarditis.
[b]Cases may present with multiple drug classes. Data are presented as number (proportion) unless noted otherwise.

Pharmacovigilance disproportionality analyses using IC have long been considered relevant toward building a case for delving deeper into associations between incriminated drugs and specific ADRs, using spontaneous reports as material. Global pharmacovigilance systems rely on spontaneous reporting systems, which provide a large volume of information and allow for the early detection of issues related to drugs or their use. While not without flaws, these systems are specifically designed to capture the information related to ADRs with dedicated and focused data collection concerning the treatment modalities. On the other hand, real-world data coming from the administrative database used for reimbursement of care, such as the French "Système national d'information inter-régimes de l'Assurance maladie" (SNIIRAM), may have larger volumes of data. However, in the latter, quality of data is driven by economic and administrative focus with lack of basic information (duration, effective start- and end-date of drug intake) and lack of information of drugs that are not reimbursed. Furthermore, as for any other measures of disproportionality, the need for caution to interpret quantitative results is paramount and IC values primarily serve to triage which drugs or drug categories require scrutiny while building case reviews[15]. Hence, the primary aim of such methods is to look at plausible drug–ADR associations, before delving deeper using combined in vitro and in vivo translational methods to assess causality[14,16].

Clinical description of cases reported in VigiBase® was consistent with previous cohorts described for each incriminated drug class. For example, antipsychotics have been mostly reported with clozapine, consistent with our data with time to onset and subsequent mortality rates similar to what has been described[6,17,18]. Salicylates were also reported in relatively young patients, with treatments indicated for chronic inflammatory bowel diseases[19]. Although inflammatory bowel disease-related myocarditis may be a confounding factor, resolution of symptoms following treatment discontinuation and recurrence following drug rechallenge tend to confirm drug-induced myocarditis[20]. Immunotherapy has been associated with myocarditis since 2015 after its first report[5]. Since the last reporting from VigiBase® in January 2018, the number of cases drastically increased from 122 to 538 in January 2020[8]. Combination immunotherapy remained associated with higher mortality. Interestingly, mortality associated with ICI-related myocarditis has decreased over time, perhaps owing to better awareness and earlier recognition and management[21]. Of all vaccines, smallpox followed by anthrax vaccines were the most reported, as described in US military personnel[22]. Finally, minocycline was associated with myocarditis, with concurrent eosinophilia (including drug reaction with eosinophilia and systemic symptoms (DRESS)) in 64% of cases. Subsequent mortality associated with minocycline-myocarditis was the highest of the studied drugs (52%), akin to that described after DRESS[23], and in a single-case metanalysis of published histologically proven eosinophilic myocarditis[6].

The spectrum of culprit drug classes in our study highlights the variety of mechanisms underlying drug-induced myocarditis. Antipsychotic antidopaminergic drugs such as clozapine are probably linked to a type-1-immunoglobulin-E-mediated hypersensitivity reaction[24], and anticholinergic blockade with high sympathetic drive responsive to beta-adrenergic blockade[25]. There have been several proposed mechanisms for salicylate-associated myocarditis: direct toxicity on the myocardium, allergic reaction mediated by immunoglobulin E, cell-mediated hypersensitivity reaction, or a humoral antibody response[19,26]. Both antipsychotic agents and salicylates are associated with eosinophilic myocarditis, and were the two classes of drugs most frequently associated with eosinophilia in this work. Immunotherapies have been associated with fulminant lymphocytic

with the highest risk of myocarditis reporting. Overall mortality risk was 10.3%, with salicylates and vaccines being associated with the lowest mortality and anticancer drugs with the highest. Of importance, two-third of the drugs (38/62, 61.3%) reported in this analysis were not labeled as associated with myocarditis by the FDA, and represent signals of suspected causality requiring further clinical reviews. While associated with inherent biases, disproportionality analyses generate signals which may allow focus on specific drug-related ADR, to further knowledge of mechanisms using preclinical platforms[14]. Finally, patients' profiles, time to onset, and co-reported adverse events varied widely depending on the drug class considered, emphasizing the polymorphism of drug-induced myocarditis.

**Table 2 Cases descriptions, by drug class, with heatmap of associated adverse drug reactions (green to red, least to most associated).**

| Drug class | Overall | Antipsychotic | Immunotherapy | Cytotoxic | Vaccine | Salicylate | p-value[a] |
|---|---|---|---|---|---|---|---|
| $N_{observed}$ | 5108 | 3108 | 538 | 190 | 790 | 340 | |
| Deaths | 524/5108 (10.3%) | 181/3108 (5.8%) | 175/538 (32.5%) | 46/190 (24.2%) | 83/790 (10.5%) | 9/340 (2.6%) | <0.0001 |
| Women patients | 1274/4866 (26.2%) | 726/2964 (24.5%) | 169/495 (34.1%) | 89/169 (52.7%) | 127/784 (16.2%) | 91/327 (27.8%) | <0.0001 |
| Age in years | 35 [25;50] [4337] | 36 [26;47] [2688] | 69 [60;74] [363] | 47 [34.5;58] [159] | 24 [20;34] [726] | 28 [20;40] [296] | <0.0001 |
| Time to onset in days | 15 [10;23] [2302] | 17 [12;24] [1196] | 33 [20;84] [139] | 9 [3;27] [69] | 10 [5;13] [588] | 17 [12;29] [138] | <0.0001 |
| *Associated cardiac events* | | | | | | | |
| Heart failure | 20.30% | 16.31% | 17.66% | 43.16% | 33.92% | 12.35% | <0.0001 |
| Pseudo-coronary | 25.60% | 22.55% | 9.29% | 16.84% | 58.35% | 13.82% | <0.0001 |
| Rhythmologic | 25.20% | 26.25% | 17.66% | 16.84% | 37.59% | 6.47% | <0.0001 |
| Pericardial | 8.80% | 6.56% | 2.04% | 10.53% | 19.49% | 14.71% | <0.0001 |
| *Concomittant adverse events* | | | | | | | |
| Abdominal (aseptic, non-hepatic) | 4.50% | 4.28% | 4.83% | 4.21% | 4.30% | 7.35% | 0.31 |
| Anaphylaxis | 0.80% | 0.48% | 0.00% | 2.11% | 0.63% | 2.06% | <0.0001 |
| Dermatologic | 2.80% | 0.93% | 4.46% | 3.16% | 6.58% | 2.94% | <0.0001 |
| Endocrine | 3.20% | 2.41% | 7.43% | 3.68% | 2.66% | 0.88% | <0.0001 |
| Eosinophilia | 4.60% | 6.08% | 0.19% | 0.00% | 1.39% | 2.06% | <0.0001 |
| Hepato-biliary | 4.80% | 3.06% | 13.38% | 5.26% | 4.94% | 1.76% | <0.0001 |
| Hydroelectrolytic imbalance | 1.80% | 1.35% | 1.67% | 1.58% | 3.67% | 0.88% | <0.0001 |
| Hyperthermia | 14.80% | 16.99% | 2.23% | 4.74% | 20.76% | 7.35% | <0.0001 |
| Infections | 9.40% | 8.24% | 4.83% | 10.53% | 17.97% | 2.65% | <0.0001 |
| Kidney disorders | 3.50% | 2.19% | 5.02% | 8.42% | 4.43% | 2.06% | <0.0001 |
| Leucopenia | 1.70% | 2.06% | 0.74% | 4.74% | 0.89% | 0.29% | <0.0001 |
| Muscular (Myositis and Myasthenia) | 10.50% | 6.31% | 29.00% | 1.05% | 20.76% | 2.06% | <0.0001 |
| Neurologic and psychiatric | 14.00% | 13.38% | 7.43% | 7.37% | 27.09% | 4.71% | <0.0001 |
| Ophtalmologic | 0.50% | 0.10% | 1.86% | 0.53% | 0.89% | 0.00% | <0.0001 |
| Osteoarticular and rheumatologic | 1.80% | 0.97% | 2.23% | 2.11% | 3.67% | 1.18% | <0.0001 |
| Overdosage | 1.40% | 1.71% | 0.56% | 0.53% | 0.13% | 0.59% | <0.0001 |
| Pulmonary (parenchymal) | 5.70% | 4.57% | 8.18% | 6.84% | 8.73% | 3.53% | <0.0001 |
| Pulmonary (pleural) | 1.40% | 1.16% | 1.86% | 0.53% | 1.77% | 1.76% | <0.0001 |
| Thrombocytopenia | 0.80% | 0.64% | 1.67% | 2.11% | 0.76% | 0.00% | <0.0001 |
| Thromboembolism (peripheral) | 1.70% | 1.29% | 2.23% | 2.63% | 2.15% | 1.47% | <0.0001 |

Detail of adverse events categorization is available as Supplementary Data. One case may be related to several drug classes. Overall calculations were performed on the cases associated with the 62 retained drugs detailed in the Methods section. Continuous data are presented as median [interquartile range][available data], categorical data as number/available data (proportion).
[a]Intergroup comparisons are using $\chi^2$ for categorical variables and Kruskal–Wallis tests for continuous variables.

myocarditis, due to immune checkpoint inhibition that is specifically mediated by T cells. Preclinical models with PD1 and CTLA4 gene deletion manifest severe myocarditis[27], while histology in human heart presenting with immunotherapy-induced myocarditis show T cells and macrophages infiltrates resembling cardiac allograft cellular rejection[5,21]. Cytotoxic agents used as antineoplastic also feature direct cytotoxicity to cardiomyocytes with myofibrillar disarray due to neuregulin 1β dysregulation, associated with mitochondrial apoptosis and free radical production mechanisms[28]. Finally, vaccines (most prominently smallpox) are associated with myocarditis, mediated by autoimmunity secondary to vaccine-mimicry of myocardium antigens[29], and more recently, activation of toll-like receptors have been more specifically discussed[30].

The evolution of drug-induced myocarditis reporting, and IC values, showed variations across time periods and medical definitions. Vaccine-related myocarditis saw an abrupt increase in reporting in 2010 in the wake of systematic vaccination of military personnel mainly against anthrax and smallpox, for which recommendations changed at the end of 2019[31]. Immune-based cancer therapies drugs which appeared after 2010 rose as one of the major drug classes leading to myocarditis[8]. Similarly, the evolution of subsequent mortality after drug-induced mortality decreased over time in most drug classes, which may reflect better awareness of this issue across all fields, with earlier management involving: drug cessation, appropriate cardiac monitoring, and in some cases, drug-toxicity reversion using drug antagonists.

Further understanding toxicity mechanisms involved in the incidence of drug-induced myocarditis may allow to propose adequate and specific treatments for this rare but potentially lethal ADR.

Once again, IC values need to be interpreted with caution. Because they can be influenced by publication bias (i.e., physicians' awareness of the drug–ADR association), high IC values for antipsychotics such as clozapine are expected, due to the general knowledge of both psychiatrists, cardiologists, and pharmacologists. This bias should be remembered when comparing IC values between drugs. Yet, a rapidly increasing IC may also reflect an increasing use of a drug category (such as ICI), hence, the increasing prevalence of ADR related to this drug. Though, despite all its limitations, IC holds better against publication bias than raw case counts. This is why assessing drug–ADR associations uniquely on individual case count (i.e., case series) is less representative than performing a full-fledged IC-based disproportionality analysis, which accounts for reported cases, and non-cases.

In this work, in addition to confirming known associations between certain drug categories and myocarditis, even if IC may be overinflated; several other signals were raised. For brevity, we will not suggest plausible mechanisms for all drugs, however, most drugs that were not previously flagged as purveyors of myocarditis belong to known drug categories (mainly antipsychotics, salicylates, and vaccines), which suggests and comforts the plausibility of some drug-class-effect.

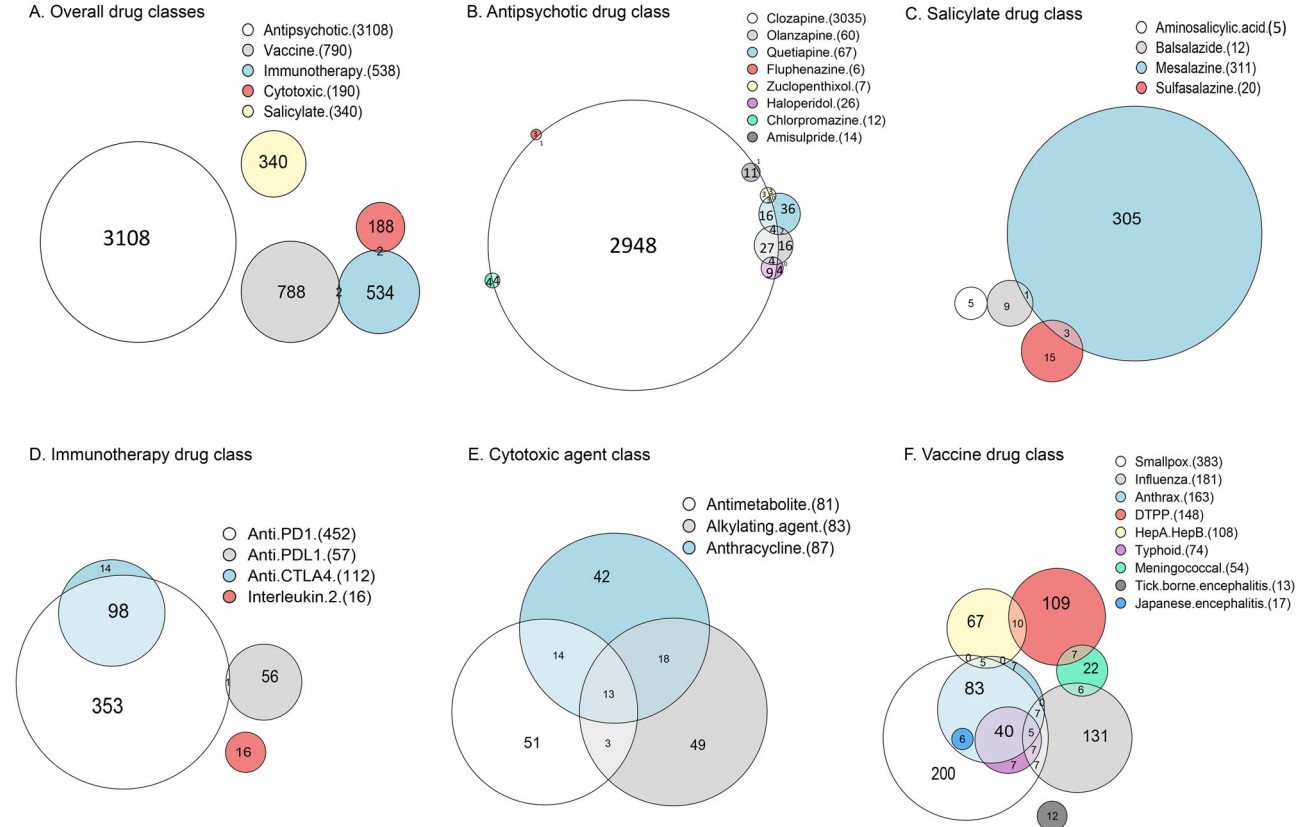

**Fig. 2 Overlap between drugs associated with myocarditis.** Overlap between drug classes in the dataset (**A**), between drug substances in the antipsychotic group (**B**) and salicylate group (**C**), between subclasses in immunotherapy (**D**) and cytotoxic (**E**), and types of vaccines in the vaccine group (**F**). DTPP diphtheria, tetanus, pertussis, and/or polio vaccine, HepA/HepB hepatitis A and/or hepatitis B vaccine. Subclasses in immunotherapy regroup anti-PD1 (cemiplimab, nivolumab, and pembrolizumab), anti-PDL1 (atezolizumab, avelumab, and durvalumab), anti-CTLA4 (ipilimumab), and interleukin-2 (aldesleukin). Subclasses in cytotoxic regroup alkylating agents (busulfan and cyclophosphamide), anthracyclines (daunorubicin, doxorubicin, epirubicin, and idarubicin), and antimetabolites (cytarabine and fluorouracil). Due to graphical limitations, some overlaps cannot be displayed. Exact combinations are presented with UpSetR plots in the Supplementary Information file.

We acknowledge several and important bias due to the nature of the pharmacovigilance database. The first being under-reporting, associated with halo bias and lack of information on the exposed population for calculation of incidence, which would require sales data from the industry. Indeed, myocarditis being mostly caused by viral infection (among other causes), the differential etiologic diagnosis may have interfered in some results[1,22]. Notably, the lack of antibiotic drugs as a full-fledged drug class may be explained by underreporting due to septic myocarditis alternative diagnosis. Drug-induced myocarditis remains a rare entity, and true incidence remains elusive, due to numerous factors. The lack of consensual definition, with multiple criteria possible: clinical symptoms, EKG modification, cardiac enzyme elevation, decrease in cardiac function assessed by either echocardiography or cMRI and exclusion of differential diagnoses; all for which details were not mentioned in VigiBase®[22]. Moreover, not being able to return to each report to ensure that an exhaustive search for etiologies and concomitant drugs intake has been carried out leads to an information bias. Moreover, preexisting cardiovascular diseases are not exhaustively collected in this data source, as only drugs and their indications are mentioned, while existing comorbidities that may not be treated cannot be reported. The likelihood of a causal relationship is not the same in all reports. As stated above, IC value comparisons lack the possibility of distinguishing between variations in reporting due to a rise in awareness or a rise in the absolute number of events. The fact that clinical presentations

yielded from this work matched that of known drug-class-associated myocarditis is reassuring. However, TTO analyses, which also rely on spontaneous reporting, may suffer from reminiscing-bias, as a case more at the chance to be related to a drug, if that drug was administered recently, as opposed to a drug administered years before. Hence, TTO may be underestimated, although a comparison between observed TTO in our study and expected TTO from known literature did not yield significant differences.

In the end, disproportionality analysis methodology allows to focus the attention of clinical physicians and to assess the plausibility of the incrimination of a drug toward a singular adverse event, i.e., myocarditis, which in the end, require validation and confirmation using translational research methods[14,16].

## Methods

**Study design.** This is a worldwide observational case–non-case cross-sectional study focusing on drug-induced myocarditis using the international pharmacovigilance database, VigiBase® (NCT03855982)[32]. VigiBase® is the WHO global ICSR deduplicated database, managed by the Uppsala Monitoring Centre (Uppsala, Sweden) (accessible at www.vigiaccess.org). It contains over 21 million ICSR received from over 130 countries since 1967 with over 25,000 drugs and vaccines. ICSR originate from different sources, such as healthcare professionals, patients, and pharmaceutical companies, and are generally notified post-marketing. ICSR includes administrative information (country, type of report, and reporter), patient data (age, sex), date of onset of reaction(s), and nature of the outcome using the latest version of Medical Dictionary for Regulatory Activities (MedDRA) terms (currently v22.1)[33]. Drug(s) involved (name, drug start and stop dates, indication, dose) are also indicated. Drugs are coded using the WHO drug dictionary and

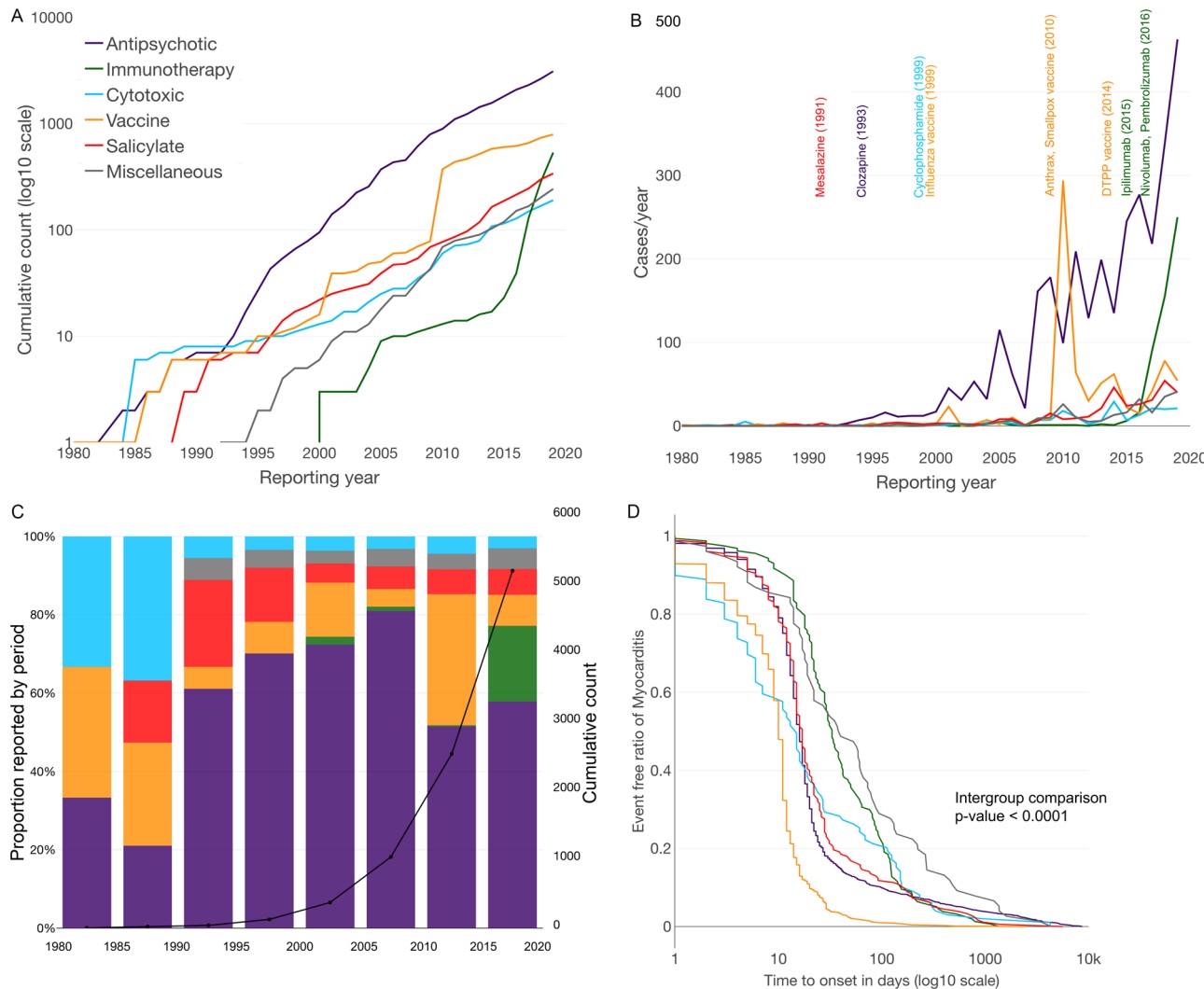

**Fig. 3 Evolution of reporting across time, by drug classes.** Evolution of cumulative number of reports per year (**A**); of cases per year, by drug class (**B**); of proportion of each drug class by half-decade (**C**); and time to onset between first treatment intake and myocarditis event (**D**). In **B**, are also mentioned all drugs with more than 80 reports over time, and corresponding year when the information component (IC) became significant with an $IC_{025} > 0$. In **D**, intergroup comparison represents the comparison between all groups, regarding the time to onset between first treatment intake and myocarditis event (using Kruskal–Wallis methods). DTPP diphteria, tetanus, pertussis and/or poliomyelitis vaccine.

categorized using the ATC classification[5]. Each event is characterized as "serious" or "non-serious" according to the WHO definition. Seriousness corresponds to death, life-threatening situations, hospitalization, hospitalization prolongation, persistent incapacity or disability, and situations judged clinically serious by the physician reporting the case. The TTO was computed as the time (in days) between the date of initiation of incriminated drug and the date of the myocarditis onset. When only the year and month were available, the date was extrapolated to the 15th of the month. TTO was considered missing otherwise. Previous association between myocarditis and a given drug was searched for in the United States FDA labels[34]. The use of VigiBase® to perform pharmacovigilance analyses is not dependent on an institutional review board approval, yet, it is conditioned on institutional access provided and approved by the Uppsala-Monitoring-Centre. Because spontaneous reporting systems are based on anonymity and the process only requires a lack of opposition from the patient, no informed consent was sought to use VigiBase® in this study.

Of note, 100 cases of drug-induced myocarditis reports, randomly extracted from the French pharmacovigilance database (part of VigiBase® with narratives accessible to our group), were retrospectively analyzed to compute the positive predictive value (true positive/(true positive+false positive)) of clinically suspected myocarditis, as defined by the ESC guidelines[35], and assess the proportion of biopsy-proven or cardiac magnetic resonance imaging-proven myocarditis. The use of confidential electronically processed anonymized patient data was approved by the French National Commission for Data Protection and Liberties (Commission Nationale de l'Informatique et des Libertés; reference: 1922081).

The computed positive predictive value was 94/100, 94%. At the time of reporting, the proportion of biopsy-proven myocarditis was 6/100, 6%, coronary involvement was excluded in 50/100, 50% and cardiac MRI was available in 37/100, 37%.

In addition, we performed a pharmacovigilance causality assessment analysis following the French method, on all drugs that were not previously described as associated with myocarditis, nor shared similar pharmacological properties as drugs that were known to be associated with myocarditis. This analysis was based on three criteria: chronological, semiological, and extrinsic accountability. Chronological criterion score corresponds to: C0: incompatible, C1: doubtful, C2: plausible, C3: probable. Semiological criterion score, which is based on semiotics, drug dechallenge or rechallenge, and existence of confounding elements (preexisting comorbidities or co-medications) corresponds to: S1: questionable, S2: plausible, S3: likely. Imputability score combines chronological and semiological criteria and corresponds to: I0: incompatible, I1: doubtful, I2: plausible, I3: likely and I4: very likely. Finally, extrinsic accountability is a bibliography score: B0: unpublished, B1: class effect, B2: widely published, and B3: expected effect (described in the product information)[13]. Results are presented in Supplementary Table C.

**Analysis in VigiBase®.** VigiBase® is a spontaneous reporting system that allows for more robust and rigorous analyses than isolated case reports or case series, due to the possibility of performing quantitative comparisons, such as disproportionality analysis (case–non-case) to identify drugs significantly associated with myocarditis[8]. We identified cases of myocarditis by searching in VigiBase® all

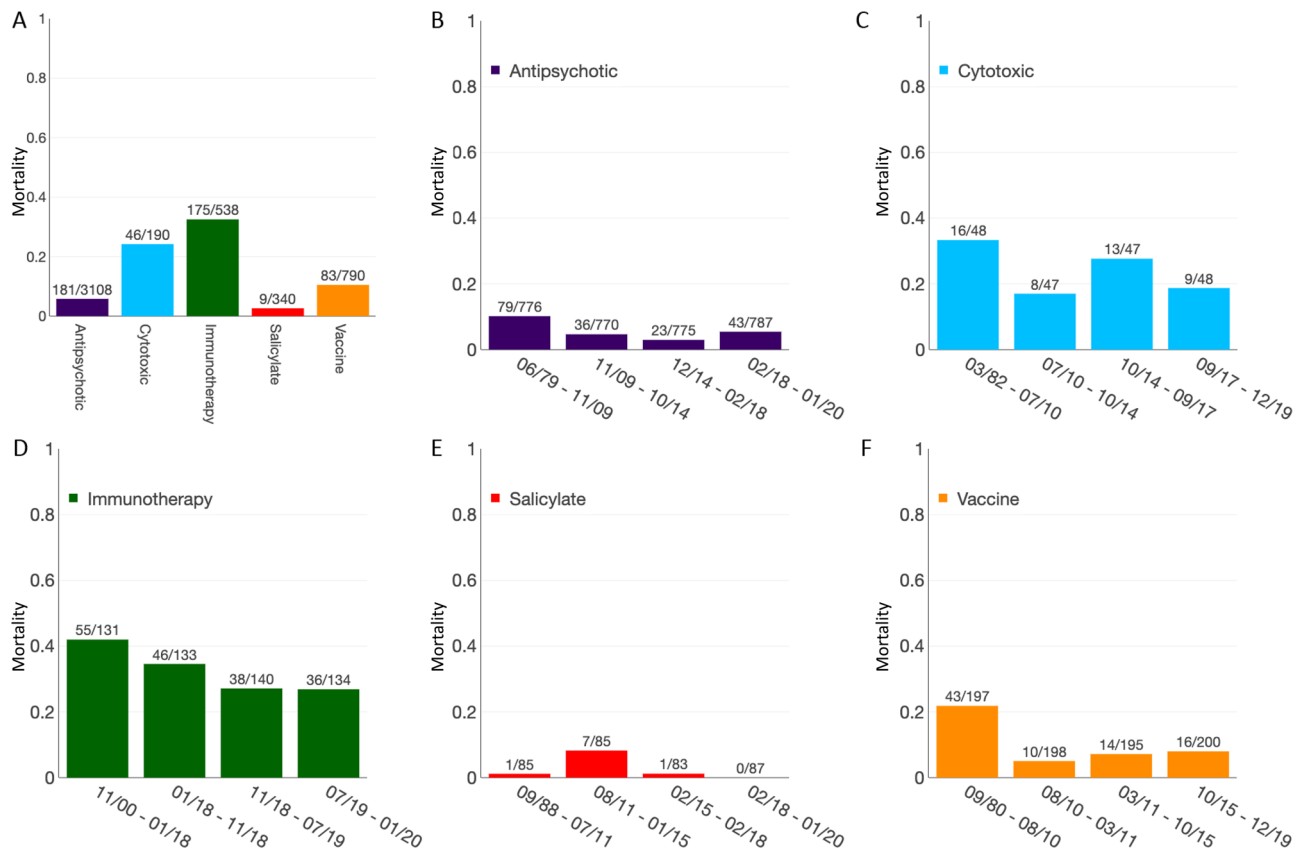

**Fig. 4 Evolution of mortality across time, by drug classes.** Mortality associated with myocarditis cases, by drug class (**A**) and mortality by quartile of time, for each drug class (**B–F**). Intergroup comparisons assessed by $\chi^2$ test were performed to assess mortality differences over time, for which results are as follow: **A** $p = 2.1e-90$ (significant), **B** $p = 2.8e-05$ (significant), **C** $p = 0.2$ (not significant), **D** $p = 0.018$ (significant), **E** $p = 0.003$ (significant), and **F** $p = 5.2e-08$ (significant).

ICSR flagged with the MedDRA preferred-term level «myocarditis» from inception to January, 12, 2020; with a drug declared as "suspect" or "interacting" with myocarditis reaction. To do so, we used the VigiLyze software with the English version 22.1 of MedDRA (Uppsala Monitoring Centre, Sweden). Disproportionality analysis compares the proportion of a selected specific ADR reported for a single drug with the proportion of the same ADR for a control group of drugs (i.e., full database with all drugs). The denominator in these analyses is the total number of ADR reported for each group of drugs. If the proportion of cases associated with a specific drug is greater than in patients without this ADR (non-cases), there is a disproportionality association (signal identification) between the ADR and the drug. In the present work, the calculated Bayesian disproportionality estimate was the IC[8]. Herein, we also performed for selected previously unknown liable drugs a sensitivity analysis excluding from full database the ICSRs in which drugs already known to be associated with myocarditis were reported.

Calculation of the IC using a Bayesian confidence propagation neural network was developed and validated by the Uppsala Monitoring Centre as a flexible, automated indicator value for disproportionate reporting that compares observed and expected drug–ADR associations to find new drug–ADR signals with identification of probability difference from the background data (full database)[36]. Probabilistic reasoning in intelligent systems (information theory) has proved to be effective for the management of large datasets, is robust in handling incomplete data, and can be used with complex variables. The information theory tool is ideal for finding drug–ADR combinations with other variables that are highly associated compared with the generality of the stored data[36]. Several examples of validation with the IC exist, showing the power of the technique to find signals sooner after drug approval than by a regulatory agency, and to avoid false positives, whereby an association between a common drug and a common ADR occurs in the database only because the drug is widely used and the ADR is frequently reported (e.g., between digoxin and rash)[36,37]. Like others, our team published several studies using VigiBase® and disproportional reporting calculation to characterize and identify new drug–ADR-associated signals, which were subsequently corroborated by preclinical mechanistic studies or prospective cohorts[8,14,,21,38,39]. This later element requires to be emphasized, as IC value should be interpreted only as means to perform clinical reviews of plausible associations and do not signify causality in any way. The IC$_{025}$ is the lower end of the 95% credibility interval for the IC. A positive value of the IC$_{025}$ is deemed significant[8,40].

The statistical formula of IC is as follows:

$$IC = \log2\left[\frac{(N_{observed} + 0.5)}{(N_{expected} + 0.5)}\right]$$

where

$$N_{expected} = \left[\frac{(N_{drug} \times N_{effect})}{N_{total}}\right]$$

$N_{expected}$ is the number of case reports expected for the drug–ADR combination. This number is computed from $N_{drug}$, $N_{effect}$, and $N_{total}$. $N_{observed}$ is the actual number of case reports for the drug–ADR combination. $N_{drug}$ is the number of case reports for the drug, regardless of ADR. For example, if overall, a drug substance X was reported in 10 cases of rash, 5 cases of fever, and 3 cases of myocarditis (all independently), $N_{drug}$ for drug substance X would be $10 + 5 + 3 = 18$. $N_{effect}$ is the number of case reports for the ADR, regardless of drug. $N_{total}$ is the total number of case reports in the database.

Drugs categorized as immunosuppressant (ATC label L04XX) were excluded to avoid indication bias. Sensitivity analysis studying the association between selected liable drugs and myocarditis by age subgroups has been performed in VigiBase using IC$_{0005} > 0$ as a significant threshold to account for multiple testings.

**Descriptive statistics**. For a description of ICSR, continuous data were reported in median [interquartile range]. MedDRA terms used to classify concurrent ADR are detailed in Supplementary Data 5. All data were available, otherwise specified. Data management was performed using Python software version 3.0 (Python software foundation, Wilmington, Delaware, USA).

**Reporting summary**. Further information on research design is available in the Nature Research Reporting Summary linked to this article.

## Data availability
Data were extracted from VigiBase®, the World Health Organization pharmacovigilance database. All data may be accessible at www.vigiaccess.org, after detailed request to the Uppsala Monitoring Center (Sweden), following privacy requirements. For more information on the process to be granted access and use of these data, please refer to their

dedicated website: www.who-umc.org. The source data that contain a dataset without any individual information related to Fig. 1 are provided with this paper. Source Data are provided with this paper.

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

## Acknowledgements

The supplied data from VigiBase® come from various sources. The likelihood of a causal relationship is not the same in all reports. The information does not represent the opinion of WHO. L.S.N. and J.-E.S. were involved in the planning, conduct, and reporting of the work described in the article. B.B.M. was involved in the conduct of the study and data management. All other authors were involved in the conduct of the study, and the editing of the manuscript. J.-E.S. is responsible for the overall content as guarantor. All authors agreed on the submitted version.

## Author contributions

L.S.N. wrote the manuscript, co-designed the study, performed analyses, and data visualization. L.T.C. and M.K. provided critical reviewing to the manuscript. C.F.-B. co-designed the study, participated to resource procurement, and provided critical reviewing to the manuscript. J.S., N.B., and E.A. provided critical reviewing to the manuscript. B.B.M., K.B., and C.D. performed data curation, data analysis, and data visualization. A.R., E.G., B.L.-V., S.E., K.B., J.J.M., and C.D. provided critical reviewing to the manuscript. J.-E.S. co-designed the study, and supervised manuscript writing, analyses, and data visualization.

## Competing interests

M.K. has received consulting fees from Sanofi, lecture fees from Bayer, a research grant from PHRC 2015 (P150921), Federation Française de Cardiologie. J.S reports during the past 2 years the following disclosures: consulting Fees or Lecture Fees or Travel Support from AstraZeneca, Bayer HealthCare SAS, Boehringer Ingelheim France, CSL Behring SA, Gilead Science, Sanofi-Aventis France, Terumo France SAS, Abbott Medical France SAS, Stockholder of Pharmaseeds. B.B.M. has no conflict of interest regarding this work, and is currently employed by Kayrros (Paris, France). E. G. received consulting fees from consulting fees from Boston Scientific, Microport and Medtronic. J.J.M. received fees from Pfizer, Novartis, Bristol-Myers Squibb, Deciphera, Audentes Pharmaceuticals, Nektar, Takeda, Ipsen, Myokardia, AstraZeneca, GlaxoSmithKline, Intrexon, and Regeneron, and is supported by R01 HL141466. J.-E. S. has participated to BMS advisory boards. The remaining authors declare no competing interests.

## Transparency declaration

The lead author (the manuscript's guarantor) affirms that the manuscript is an honest, accurate, and transparent account of the study being reported; that no important aspects of the study have been omitted; and that any discrepancies from the study as planned (and, if relevant, registered) have been explained.
