## [Peer Review File · Nature Communications]

REVIEWER COMMENTS

Reviewer #1 (Remarks to the Author):

This pharmacovigilance study of Nguyen et al. is worldwide observational case-non-case cross-sectional study focusing on drug-induced myocarditis using an international pharmacovigilance database. From VigiBase® inception (1967) to January 2020, 6823 ICSR of suspected drug-induced 150 myocarditis were reported from a total of 21,185,309 ICSR in the full database, and from 47 151 countries. They identified 5108 ICSR for drugs for which reporting of myocarditis as an adverse event 152 was significantly increased compared to all other drugs in VigiBase®. They identified a list of 62 drugs associated with myocarditis, comprising 66 five major drug classes. The study illustrates the 67 diversity of presentations of drug-induced myocarditis.

This substantial manuscript is excellent an of clinical relevance. The study design is interesting in principle, however, have one major remark on the manuscript: It is not clear what the diagnosis myocarditis is based on. It is absolutely apparent that this is a reliable diagnosis or whether it was just a matter of suspicion, which gave rise to the term "myocarditis".

The manuscript must be revised with a detailed breakdown of the diagnosis (suspected clinical diagnosis, post-mortem analysis, imaging, endomyocardial biopsy).

Reviewer #2 (Remarks to the Author):

The authors describe a data mining analysis of drug-induced myocarditis through spontaneous reports of suspected adverse drugs reactions reported to the WHO database, using disproportionality analysis (specifically the Information Component).The authors also examine reported time to onset.

This is of course an important topic and I am sure there are important findings that can be gleaned from this analysis, however further work is needed to get to that point, and in its current form the manuscript overinterprets IC values on Vigibase data.

Major comments:

The Information Component is a now long established routine approach for screening spontaneous reports, standard good practice is that quantitative statistical alerts, as for other measures of disproportionality, should be interpreted with great caution and act primarily as a triage for where to focus clinical case review (see eg. Wisniewski, A.F et al, 2016. Good signal detection practices: evidence from IMI PROTECT. *Drug safety*, 39(6), pp.469-490.).

So the authors intention to look to identify drugs significantly associated with myocarditis, and describe their relative prevalence over time, should be addressed with more appropriate data e.g. Real World Data (EHR, Claims, registries). If for some reason the authors' consider this infeasible a strong rationale for spontaneous reports, despite their many limitations, being the best available data source should be provided. This is not due to limitations with the quantitative measures themselves, but that spontaneous reports have long established limitations (ref e.g. Rawlins, M.D., 1988. Spontaneous reporting of adverse drug reactions. II: Uses. *British journal of clinical pharmacology*, 26(1), p.7.) that mean that quantitative associations and prevalence estimation cannot be done with spontaneous reports because of underreporting and differential reporting due to e.g. publication bias.

Of importance, publication bias/knowledge of an ADR is well established to increase reporting rates and inflate disproportionality scores including IC values.

Specifically therefore they should differentiate in their disproportionality analyses between known and unknown ADRs, ie analyse separately currently unknown or potential causes of myocarditis from established causes. The latter classes of drugs will have far less underreporting and will also potentially include cases where the product is erroneously considered the culprit of the adverse events, because it is such a well established ADR in general.

Myocarditis specific analyses of the WHO database has been conducted previously using Information Component analysis, although not to my knowledge considering all myocarditis reporting, focussing e.g. on antipsychotics (refs Moslehi JJ et al, 2018. Increased reporting of fatal immune checkpoint inhibitor-associated myocarditis. *The Lancet*, 391(10124), p.933. Coulter DM et al 2001. Antipsychotic drugs and heart muscle disorder in international pharmacovigilance: data mining study. *Bmj*, 322(7296), pp.1207-1209 and Hagg S et al 2001. Myocarditis related to clozapine treatment. *Journal of clinical psychopharmacology*, 21(4), pp.382-388.). It would be relevant to understand how antipsychotic results differ in this more recent manuscript and also differences in analysis strategy.

Coulter et al make the point that in the WHO database myocarditis that has occurred may be mistakenly diagnosed and/or reported with other terms e.g. cardiomyopathy. The authors should discuss the risk of missing myocarditis cases from reporting other terms, and their rationale for choice of meddra code. Differential choice of codes over time or between medicinal products may

explain - or hide - differences and perceived trends. This needs to be discussed and a sensitivity analysis with different meddra code selection would strengthen the study.

Of note Information Component and other measures of disproportionality are more robust to class wide publication bias than raw case counts, and the authors' might want to make this point more clearly when discussing e.g. case counts. Confounding/effect modification can be problematic for spontaneous report analysis so subgroup analysis considering e.g. age group specific IC scores may also be informative. If no such subgroup analysis is included in a publication of this sort I would expect some justification to be provided.

The authors also look to report the clinical features of all reported drug-induced myocarditis according to drug classes, which is an area where spontaneous reports can provide useful insights, as well as identifying potential new signals. More extensive evaluation should be conducted.

Re the TTO analysis, I think this is an important component of the manuscript. However the authors need to be careful in the assumptions they make around missing data being assumed to be the middle of the month and also missing data may well be systematically different to recorded data. With spontaneous reports one would expect suspected causality to be triggered if onset of adverse event is nearer to drug or vaccine exposure – and this likely bias should be stated explicitly and the anticipated likely impact on the results discussed.

Specific comments:

“A positive value of the IC025 is deemed significant”, is not statistically significant as in the same sense if it were a RCT or epidemiological study – I think important to write in full that means quantitative significantly more reporting than one would expect in terms of general reporting in the database ie it is relative to a background of other reports. The reason this is so important is that differential reporting patterns will affect the disproportionality (particularly for labelled events) and therefore quantitative scores need to be interpreted with caution. Potential issues such as masking are essential to consider. Disproportionality scores are predictive of signals, but clinical review is an essential follow up step.

“Drugs categorized as immunosuppressant (ATC label L04XX) were excluded to avoid indication bias.” Care is needed here of course – is there a risk of patients who had been co-administered L04XX drugs had that information missing on the reports? Perhaps minimal chance, but useful to mention.

Comparisons between the classes, e.g. in terms of age distributions can be interesting, but again caution is advised not to make too strong inference – differential chance and type of reporting (or missing data) could explain the differences that are seen, and the authors need to make the alternative explanations clear and motivate why they believe the differences represent genuine

differences. TTO onset differences will be systematic between drug classes depending on the setting of drug prescribing/administration and the likelihood of careful follow up as reporting patterns are likely to vary by type and frequency of healthcare encounter. Similarly the probability of follow up information on mortality will vary greatly in a non-random way over time, between product, country etc.

The following statement is incorrect:

“Of importance, two third of the drugs (38/62, 61.3%) reported in this analysis were not labeled as associated with myocarditis by the Food and Drug Administration and thus, represent new signals” they do not represent signals as used by WHO/FDA/EMA – they represent statistical alerts that require detailed clinical review to be considered safety signals.

Disproportionality analyses do not “generate signals” they generate statistical alerts that require clinical review on a case level to be subsequently considered signals. The authors could use the term “signals of disproportional reporting” but would then have to explicitly mention the difference to “signals of suspected causality”.

Similarly the authors conclude:” This study identified 62 drugs associated with myocarditis, 38 of which were not previously reported in FDA labels. They were grouped into: antineoplastic cytotoxics and immunotherapies,antipsychotics, salicylates, and vaccines. The time between treatment and myocarditis onset, presenting clinical features, subsequent mortality and patients’ profile varied significantly between

the 5 main drug classes.” However in a large database like Vigibase there are least some reports on virtually all possible drug/vaccine-AE reports: some with well-founded suspicion some only ill-founded. This conclusion is as currently phrased at best therefore not interesting and if ‘association’ is taken to allude to suspected causality might be misleading -recommend the authors restate to make their novel contribution more clear.

Recommendations

1. Subdivide the Myocarditis reports by drug into known and unknown.
2. Focus all following analyses on the unknown subset, including the IC analysis
3. Consider subgroup analysis if confounding/effect modification is suspected.
4. Conduct detailed clinical review of potential emerging safety signals

5. For emerging potential signal consider widening MEDDRA code selection to consider myocarditis cases erroneously captured with less specific codes

Andrew Bate

Answer to reviewers – R1

Reviewer #1 (Remarks to the Author):

This pharmacovigilance study of Nguyen et al. is worldwide observational case-non-case cross-sectional study focusing on drug-induced myocarditis using an international pharmacovigilance database. From VigiBase® inception (1967) to January 2020, 6823 ICSR of suspected drug-induced myocarditis were reported from a total of 21,185,309 ICSR in the full database, and from 47 countries. They identified 5108 ICSR for drugs for which reporting of myocarditis as an adverse event was significantly increased compared to all other drugs in VigiBase®. They identified a list of 62 drugs associated with myocarditis, comprising five major drug classes. The study illustrates the diversity of presentations of drug-induced myocarditis.

This substantial manuscript is excellent and of clinical relevance. The study design is interesting in principle, however, have one major remark on the manuscript: It is not clear what the diagnosis myocarditis is based on. It is absolutely apparent that this is a reliable diagnosis or whether it was just a matter of suspicion, which gave rise to the term "myocarditis".

The manuscript must be revised with a detailed breakdown of the diagnosis (suspected clinical diagnosis, post-mortem analysis, imaging, endomyocardial biopsy).

Answer: We thank the reviewer for reading our work, his/her encouraging comments and suggestions to ameliorate this work. In this database, the diagnosis of myocarditis suspected to be drug-induced was overwhelmingly reported by healthcare professionals (~97%) considering this diagnosis as finally retained (using the MedDRA dictionary). The individual details of exams performed and how myocarditis diagnosis was established can be found in the narratives of each reports, but these narratives are in general not accessible to all researchers since those data might be identifying (access to those narratives are variable depending on reporting country policies, and written in each country language). Only anonymized English-translated synthetic standardized reports classified by MedDRA terms are accessible to international pharmacovigilance researchers feeding this database. Though, to answer the reviewer's concern at best of our ability (even though biased because a-posteriori monitored reports were only from one country), we were able to access to the narratives of myocarditis cases arising from France. We analyzed and monitored the narratives of 100 random reports of suspected drug-induced myocarditis, extracted by Kevin Bihan (now an author of the study) from the French pharmacovigilance database, part of VigiBase®.

We computed the positive predictive value towards clinically-suspected myocarditis (as defined by the ESC, which require a combination between symptoms, ECG modification, troponin elevation, echocardiographic elements, cardiac magnetic resonance imaging and biopsy elements). Our results comforted the fact that reports of suspected myocarditis fulfilled indeed the ESC definition of clinically-suspected myocarditis, and that more than half were assessed for coronary involvement, and found valid for myocarditis.

We added a specific part in the Methods section as follow:

"Of note, 100 cases of drug-induced myocarditis reports, randomly extracted from the French pharmacovigilance database (part of VigiBase® with narratives accessible to our group), were retrospectively analyzed to compute the positive predictive value (true positive/(true positive+false positive)) of clinically-suspected myocarditis, as defined by the ESC guidelines, and assess the proportion of biopsy-proven or cardiac magnetic resonance imaging-proven myocarditis. The positive predictive value was 94/100, 94%. At the time of reporting, the proportion of biopsy-proven myocarditis was 6/100, 6%, coronary involvement was excluded in 50/99, 50.5% and cardiac MRI was available in 37/100, 37%."

Reviewer #2 (Remarks to the Author):

The authors describe a data mining analysis of drug-induced myocarditis through spontaneous reports of suspected adverse drug reactions reported to the WHO database, using disproportionality analysis (specifically the Information Component). The authors also examine reported time to onset. This is of course an important topic and I am sure there are important findings that can be gleaned from this analysis, however further work is needed to get to that point, and in its current form the manuscript overinterprets IC values on Vigibase data.

Answer: Dear Mr Bate,

First of all, we heartfully thank you, for your time and insights, regarding our manuscript. We are very grateful that you took the time to review our work, and feel honored that you did, as disproportionality analyses have, in our opinion, been a very important tool towards allowing us to build expertise, mainly in the cardiovascular field, but also other domains.

We perfectly understand the need for caution when interpreting IC signals, which is why, we tried to tone-down the message which was conveyed and emphasize the need for i) caution when interpreting these results and ii) confirmation by using other means, such as translational research (which we spearheaded for several drugs, including antineoplastic agents (ibrutinib)(Salem, Circulation 2018 and 2020), androgen deprivation therapy (Salem Circulation 2019) and immunotherapy (Wei Cancer Discov 2021)). In the end, this work primarily aimed to establish an instant overview of very broad drug categories plausibly associated with myocarditis adverse events, and suggest common mechanisms, when possible, or relate to very incomplete knowledge which require further investigations, to raise awareness in treating physicians as well as pharmacovigilant colleagues.

In this revised version, we further toned-down our findings, and explained how Information Component was affected by over-reporting, i.e. in known drugs.

Major comments:

The Information Component is a now long established routine approach for screening spontaneous reports, standard good practice is that quantitative statistical alerts, as for other measures of disproportionality, should be interpreted with great caution and act primarily as a triage for where to focus clinical case review (see eg. Wisniewski, A.F et al, 2016. Good signal detection practices: evidence from IMI PROTECT. Drug safety, 39(6), pp.469-490.).

Answer: We perfectly agree that IC should be interpreted with caution and added several paragraphs to the manuscript to that effect. At the beginning of the Discussion section, we added Wisniewski reference in a paragraph to state the primary goal of IC analyses:

Added to Discussion: *"Pharmacovigilance disproportionality analyses using IC have long been considered relevant towards building case for delving deeper into associations between incriminated drugs and specific adverse drug reactions, using spontaneous reports as material. As for any other measures of disproportionality, the need for caution to interpret quantitative results is paramount and IC numbers primarily serve to triage which drugs or drug categories require scrutiny while building case reviews.(18) Hence, the primary aim of*

such methods is to look at plausible drug-ADR associations, before delving deeper using combined in-vitro and in-vivo translational methods to assess causality.(17)

So the authors intention to look to identify drugs significantly associated with myocarditis, and describe their relative prevalence over time, should be addressed with more appropriate data e.g. Real World Data (EHR, Claims, registries).

Answer: Unfortunately, to date, there are no exhaustive registries focusing on myocarditis due to drugs. Indeed, no specific code exists to identify the specificity of this disease in EHR, and we only have case reports and series at our disposal, as of now.

If for some reason the authors' consider this infeasible a strong rationale for spontaneous reports, despite their many limitations, being the best available data source should be provided. This is not due to limitations with the quantitative measures themselves, but that spontaneous reports have long established limitations (ref e.g. Rawlins, M.D., 1988. Spontaneous reporting of adverse drug reactions. II: Uses. British journal of clinical pharmacology, 26(1), p.7.) that mean that quantitative associations and prevalence estimation cannot be done with spontaneous reports because of underreporting and differential reporting due to e.g. publication bias. Of importance, publication bias/knowledge of an ADR is well established to increase reporting rates and inflate disproportionality scores including IC values. Specifically therefore they should differentiate in their disproportionality analyses between known and unknown ADRs, ie analyse separately currently unknown or potential causes of myocarditis from established causes. The latter classes of drugs will have far less underreporting and will also potentially include cases where the product is erroneously considered the culprit of the adverse events, because it is such a well established ADR in general.

Answer: We perfectly agree that known drugs-ADR association tend to self-inflate IC values, which is why we now insisted more on this point in the Discussion section. Furthermore, we performed a sensitivity analysis excluding from background database all reports including drugs already identified as being associated within myocarditis among the 62 liable drugs identified herein.

Added to Methods: *“Herein, we also performed for selected previously unknown liable drugs a sensitivity analysis excluding from full database the ICSRs in which drugs already known to be associated with myocarditis were reported.”*

Added to Results: *A sensitivity analysis by age subgroups; and after exclusion from full database of ICSR including drugs already identified as at known risk of myocarditis in FDA's labels among these 62 drugs is available in Supplementary-Table-C.*

Added to Discussion: *“Once again, IC values need to be interpreted with caution. Because they can be influenced by publication bias (i.e. physicians' awareness of the drug-ADR association), high IC values for antipsychotics such as clozapine are expected, due to the general knowledge of both psychiatrists, cardiologists and pharmacologists. This bias should be remembered when comparing IC values between drugs. Yet, a rapidly increasing IC may also reflect an increasing use of a drug category (such as ICI), hence, the increasing prevalence of ADR related to this drug. Though, despite all its limitations, IC holds better against publication bias, than raw case count. This is why assessing drug-ADR associations uniquely on individual case count (i.e. case series), is less representative than performing a full-fledged IC-based disproportionality analysis, which accounts for reported cases, and non-cases.”*

Myocarditis specific analyses of the WHO database has been conducted previously using Information Component analysis, although not to my knowledge considering all myocarditis reporting, focussing e.g. on antipsychotics (refs Coulter DM et al 2001. Antipsychotic drugs and heart muscle disorder in international pharmacovigilance: data mining study. *Bmj*, 322(7296), pp.1207-1209 and Hagg S et al 2001. Myocarditis related to clozapine treatment. *Journal of clinical psychopharmacology*, 21(4), pp.382-388.). It would be relevant to understand how antipsychotic results differ in this more recent manuscript and also differences in analysis strategy.

Answer: Indeed, to our knowledge, our work is first to provide a global analysis of suspected drugs associated with myocarditis versus previous studies focusing on one drug, or one class of drugs at most. As stated, like others, our team previously reported specific drug categories associated with myocarditis (mostly immune checkpoint inhibitors), but the present work aimed to compare types of reported presentations, outcomes of suspected drug-induced myocarditis as a function of molecules and drug classes. For example, in our work with ICI-induced myocarditis, we previously showed for the first time that concomitant reported ADR prominently featured neuromuscular symptoms such as myasthenia-gravis and myositis, at the time of myocarditis adverse event or briefly preceding it. In hindsight, that reflected the selective musculo-skeletal targeting of the disease (all muscular cells being destroyed), and allowed for better screening of such patients, as well as leading to specific treatments.

It is our hope that, using similar methods in other fields, early recognition of myocarditis may be possible and lead to treatment, by using concomitant reported ADR, such as fever (antipsychotics), or eosinophilia (minocycline).

Specifically, regarding antipsychotics results, we added in the Discussion how the increase in IC value could be related to an expected (yet, impossible to confirm), rise due to general awareness of the disease.

Added to Discussion: *"Once again, IC values need to be interpreted with caution. Because they can be influenced by publication bias (i.e. physicians' awareness of the drug-ADR association), high IC values for antipsychotics such as clozapine are expected, due to the general knowledge of both psychiatrists, cardiologists and pharmacologists. This bias should be remembered when comparing IC values between drugs. Yet, a rapidly increasing IC may also reflect an increasing use of a drug category (such as ICI), hence, the increasing prevalence of ADR related to this drug."*

Coulter et al make the point that in the WHO database myocarditis that has occurred may be mistakenly diagnosed and/or reported with other terms e.g. cardiomyopathy. The authors should discuss the risk of missing myocarditis cases from reporting other terms, and their rationale for choice of meddra code. Differential choice of codes over time or between medicinal products may explain - or hide - differences and perceived trends. This needs to be discussed and a sensitivity analysis with different meddra code selection would strengthen the study.

Answer: We have now studied the positive predictive value (PPV) for detecting clinically-suspected myocarditis (as defined by the European Society of Cardiology) using the medDRA preferred terms "myocarditis" in the narrative reports of the French pharmacovigilance database (part of VigiBase to which we have an authorized access to narratives). We compared it to the term "cardiomyopathy" in the same database.

We found that “myocarditis” term yielded a much higher PPV (assessed on narratives of 100 random reports selected for each term) than “cardiomyopathy” (94% vs. 1%, respectively) supporting our query strategy, and suggesting a minimal added value for using cardiomyopathy, as an alternative term.

We feel adding this element may confuse the readers as the preferred term regarding myocarditis, hence, we only added the results regarding myocarditis adjudication in the revised manuscript.

We added a specific part in the Methods section as follow:

“Of note, 100 cases of drug-induced myocarditis reports, randomly extracted from the French pharmacovigilance database (part of VigiBase® with narratives accessible to our group), were retrospectively analyzed to compute the positive predictive value (true positive/(true positive+false positive)) of clinically-suspected myocarditis, as defined by the ESC guidelines, and assess the proportion of biopsy-proven or cardiac magnetic resonance imaging-proven myocarditis. The positive predictive value was 94/100, 94%. At the time of reporting, the proportion of biopsy-proven myocarditis was 6/100, 6%, coronary involvement was excluded in 50/99, 50.5% and cardiac MRI was available in 37/100, 37%.”

Of note Information Component and other measures of disproportionality are more robust to class wide publication bias than raw case counts, and the authors’ might want to make this point more clearly when discussing e.g. case counts.

Answer: As mentioned above, we added a specific paragraph in the Discussion section to underline these elements.

Added to Discussion: *“Though, despite all its limitations, IC holds better against publication bias, than raw case count. This is why assessing drug-ADR associations uniquely on individual case count (i.e. case series), is less representative than performing a full-fledged IC-based disproportionality analysis, which accounts for reported cases, and non-cases.”*

Confounding/effect modification can be problematic for spontaneous report analysis so subgroup analysis considering e.g. age group specific IC scores may also be informative. If no such subgroup analysis is included in a publication of this sort I would expect some justification to be provided.

Answer: As requested, we have performed a sensitivity analysis for the 62 liable drugs of IC values by age subgroups (0-27 days, 28days-23months, 2-11, 12-17, 18-44, 45-64, 65-74, ≥75 years old). Results are now displayed in Supplementary Table C.

Added in Methods: *“Sensitivity analysis studying association between selected liable drugs and myocarditis by age subgroups have been performed in VigiBase using $IC_{0005}>0$ as significant threshold to account for multiple testings.”*

Added in Results: *A sensitivity analysis by age subgroups; and after exclusion from full database of ICSR including drugs already identified as at known risk of myocarditis in FDA’s labels among these 62 drugs is available in Supplementary-Table-C.*

Added **Supplementary** **Table** **C** **(see** **after)**

Supplementary Table C. Cases description, by drug substance and relevant subgroups analysis.

Substance	WHO ATC code	N substance	N observed	IC (IC ₀₂₅) vs. full database	N country	IC (IC ₀₂₅) vs. full database excluding reports with drugs already flagged for myocarditis*	Previously reported*	IC (IC ₀₀₀₅) vs. full database by subgroups of age (years)				
								0-17	18-44	45-64	65-74	>75
Antipsychotic												
Amisulpride	N05AL05	5917	14	2.6 (1.7)	4	NA	No	2.2 (-2.8)	1.9 (0.2)	NA	NA	NA
Chlorpromazine	N05AA01	11224	12	1.6 (0.7)	4	NA	Yes	NA	0.3 (-2.5)	2.5 (0.2)	1.9 (-3.1)	NA
Clozapine	N05AH02	145208	3035	6.0 (5.9)	32	NA	Yes	5.9 (5.2)	5.1 (5.0)	5.8 (5.6)	5.2 (4.6)	4.6 (3.5)
Fluphenazine	N05AB02	4363	6	1.8 (0.4)	5	1.9 (-0.2)	No	1.5 (-6.1)	1.1 (-2.2)	2.4 (-0.8)	NA	NA
Haloperidol	N05AD01	30583	26	1.3 (0.7)	6	1.1 (-0.2)	No	NA	0.7 (-0.7)	2.2 (0.3)	1.5 (-3.5)	NA
Olanzapine	N05AH03	62298	60	1.5 (1.2)	13	1.4 (0.7)	No	2.2 (-0.6)	1.0 (0.1)	1.5 (-0.04)	0.4 (-7.3)	2.0 (-1.8)
Quetiapine	N05AH04	76947	67	1.4 (1.0)	12	NA	Yes	0.2 (-7.4)	1.3 (0.6)	0.8 (-0.8)	0.1 (-7.5)	NA
Zuclopenthixol	N05AF05	3803	7	2.1 (0.9)	3	2.03 (-0.02)	No	1.5 (-6.1)	0.8 (-3.0)	2.3 (-1.6)	NA	NA
Immunotherapy												
Aldesleukin	L03AC01	1575	16	4.0 (3.2)	2	NA	Yes	NA	2.6 (-0.6)	3.4 (1.2)	NA	NA
Atezolizumab	L01XC32	4762	33	4.0 (3.5)	9	NA	Yes	NA	NA	3.8 (2.0)	3.7 (1.8)	3.3 (0.5)
Avelumab	L01XC31	696	8	3.5 (2.4)	3	NA	Yes	NA	1.5 (-6.1)	NA	2.7 (-1.1)	2.3 (-2.7)
Cemiplimab	L01XC33	269	3	2.6 (0.5)	1	NA	Yes	NA	NA	NA	NA	NA
Durvalumab	L01XC28	2611	16	3.6 (2.8)	7	NA	Yes	NA	NA	1.9 (-3.1)	2.8 (-0.4)	3.6 (1.1)
Ipilimumab	L01XC11	18436	112	4.1 (3.8)	16	NA	Yes	NA	2.5 (0.6)	3.2 (1.9)	5.1 (4.2)	4.4 (2.9)
Nivolumab	L01XC17	40718	300	4.5 (4.3)	22	NA	Yes	NA	2.9 (1.4)	4.1 (3.4)	5.5 (4.9)	5.7 (4.9)
Pembrolizumab	L01XC18	21495	149	4.3 (4.1)	25	NA	Yes	NA	2.4 (-0.1)	4.0 (3.0)	5.1 (4.3)	4.9 (3.8)
Cytotoxic												
Cyclophosphamide	L01AA01	93027	81	1.4 (1.1)	14	NA	Yes	1.6 (-0.9)	1.1 (0.2)	1.1 (-0.04)	1.5 (-0.8)	0.6 (-7.0)
Busulfan	L01AB01	5189	6	1.6 (0.2)	3	NA	Yes	NA	1.8 (-1.4)	NA	1.5 (-6.1)	NA
Cytarabine	L01BC01	29757	25	1.3 (0.7)	4	2.1 (1.1)	No	0.7 (-3.2)	-0.1 (-2.8)	2.0 (0.3)	2.1 (-1.2)	NA
Fluorouracil	L01BC02	73856	56	1.2 (0.8)	10	2.7 (2.3)	No	2.1 (-2.8)	0.9 (-0.8)	1.7 (0.7)	-1.3 (-8.9)	2.4 (-0.06)
Doxorubicin	L01DB01	74121	44	0.9 (0.4)	16	NA	Yes	0.7 (-4.3)	-0.2 (-2.0)	0.8 (-0.6)	1.3 (-1.5)	1.5 (-3.5)

Daunorubicin	L01DB02	7421	12	2.1 (1.2)	4	3.0 (1.9)	No	0.5 (-7.1)	1.7 (-1.1)	2.3 (-0.9)	1.3 (-6.4)	NA
Epirubicin	L01DB03	18370	12	1.0 (0.0)	3	NA	No	NA	0.9 (-1.6)	0.1 (-3.8)	NA	NA
Idarubicin	L01DB06	4004	22	3.6 (3.0)	4	NA	Yes	2.1 (-2.9)	2.8 (0.9)	2.6 (-0.2)	2.0 (-2.9)	NA
Vaccine												
Anthrax v.	J07AC01	9588	163	5.5 (5.3)	1	2.9 (1.9)	No	NA	4.4 (4.0)	1.6 (-3.3)	NA	NA
Meningococcal v.	J07AH10	99170	54	0.7 (0.3)	12	1.6 (1.1)	No	0.2 (-1.0)	1.4 (0.3)	0.6 (-7.0)	NA	NA
Diphtheria; Pertussis; Tet v.	J07AJ52	204958	108	0.7 (0.4)	11	NA	Yes	0.4 (-0.3)	1.1 (0.1)	1.0 (-1.5)	NA	1.3 (-6.3)
Diphtheria; Tetanus v.	J07AM	46377	24	0.7 (0.0)	9	NA	Yes	0.0 (-2.8)	0.6 (-0.8)	0.6 (-2.7)	NA	NA
Typhoid v.	J07AP	14052	74	3.9 (3.5)	4	2.1 (0.8)	No	NA	3.3 (2.7)	0.3 (-7.4)	NA	NA
Tick-borne encephalitis v.	J07BA01	10449	13	1.8 (0.9)	3	2.9 (2.0)	No	0.9 (-4.0)	1.8 (-0.2)	1.3 (-3.7)	NA	NA
Japanese encephalitis v.	J07BA02	4147	17	3.2 (2.5)	3	3.0 (1.7)	No	2.6 (-0.2)	2.4 (0.4)	1.3 (-6.3)	NA	NA
Influenza v.	J07BB02	231102	181	1.3 (1.0)	21	2.4 (2.2)	No	0.6 (-0.7)	0.9 (0.4)	0.6 (-0.4)	2.0 (1.0)	2.4 (0.9)
Hepatitis b v.	J07BC01	100233	71	1.1 (0.8)	11	2.4 (1.9)	No	1.4 (0.4)	-0.01 (-1.1)	0.8 (-1.7)	1.2 (-6.4)	NA
Hepatitis a v.	J07BC02	42781	30	1.1 (0.5)	6	1.4 (0.4)	No	0.6 (-1.1)	0.9 (-0.7)	0.8 (-4.2)	NA	NA
Hepatitis a ;Hepatitis b v.	J07BC20	10195	13	1.8 (0.9)	5	3.2 (2.3)	No	0.9 (-6.7)	1.2 (-0.6)	NA	NA	NA
Smallpox vaccine	J07BX01	6767	383	7.2 (7.0)	4	NA	Yes	NA	6.1 (5.9)	2.9 (0.1)	NA	NA
Diphtheria; Polio; Tet v.	J07CA01	6729	18	2.8 (2.0)	8	NA	Yes	2.2 (-0.3)	2.7 (1.0)	0.9 (-6.7)	NA	NA
Salicylate												
Aminosalicylic acid	A07EC	2501	5	2.1 (0.5)	3	2.8 (1.3)	No	NA	1.4 (-1.8)	1.1 (-6.5)	NA	NA
Sulfasalazine	A07EC01	24139	20	1.3 (0.6)	8	NA	Yes	NA	1.4 (-0.02)	-0.3 (-5.3)	1.1 (-3.9)	NA
Mesalazine	A07EC02	17129	311	5.7 (5.5)	26	NA	Yes	5.2 (4.2)	5.4 (5.0)	5.0 (4.2)	3.3 (1.0)	2.1 (-2.9)
Balsalazide	A07EC04	482	12	4.3 (3.3)	4	4.1 (3.0)	No	NA	4.1 (2.3)	1.5 (-6.1)	1.6 (-6.1)	NA
Miscellaneous												
Stanozolol	A14AA02	442	3	2.4 (0.4)	2	2.7 (0.6)	No	NA	2.5 (-1.4)	NA	NA	NA
Norepinephrine	C01CA03	1863	4	2.0 (0.3)	2	2.7 (0.9)	No	NA	1.7 (-3.3)	NA	NA	2.2 (-2.8)
Dobutamine	C01CA07	2495	5	2.1 (0.5)	3	2.5 (0.8)	No	NA	NA	1.2 (-6.4)	NA	NA
Milrinone	C01CE02	978	5	2.7 (1.2)	2	2.9 (1.2)	No	NA	NA	NA	NA	NA
Acitretin	D05BB02	3679	5	1.7 (0.2)	5	2.6 (1.1)	No	NA	1.8 (-2.1)	0.8 (-6.8)	NA	NA
Liothyronine	H03AA02	2484	5	2.1 (0.5)	3	2.8 (1.3)	No	NA	2.5 (-0.8)	NA	NA	NA
Minocycline	J01AA08	13162	25	2.4 (1.8)	7	NA	Yes	2.6 (0.1)	1.2 (-0.5)	2.3 (-0.5)	2.0 (-3.0)	1.4 (-6.2)
Garenoxacin	J01MA19	969	3	2.1 (0.1)	1	2.0 (-0.6)	No	NA	0.1 (-6.6)	1.4 (-6.2)	1.5 (-6.1)	NA
Daptomycin	J01XX09	6462	8	1.7 (0.5)	2	4.2 (3.5)	No	NA	0.6 (-7.1)	4.4 (3.1)	1.1 (-6.5)	NA
Rituximab	L01XC02	77766	43	0.8 (0.3)	17	2.1 (1.6)	No	0.6 (-7.0)	0.4 (-1.4)	1.1 (-0.3)	0.8 (-1.9)	1.3 (-2.5)
Trastuzumab	L01XC03	31989	27	1.3 (0.7)	12	1.8 (0.9)	No	NA	1.3 (-0.6)	0.9 (-1.4)	NA	1.9 (-3.1)

Arsenic trioxide	L01XX27	1857	4	2.0 (0.3)	3	2.7 (0.9)	No	2.2 (-2.8)	NA	NA	NA	NA
Vemurafenib	L01XE15	8971	8	1.3 (0.1)	3	1.4 (-0.6)	No	NA	1.9 (-1.3)	1.6 (-2.2)	1.1 (-6.6)	NA
Cobimetinib	L01XE38	1988	10	3.2 (2.2)	3	NA	Yes	NA	NA	3.5 (1.2)	2.6 (-1.2)	NA
Tretinoin	L01XX14	6044	20	3.1 (2.4)	6	3.3 (2.4)	No	2.8 (-0.4)	2.6 (1.0)	NA	1.3 (-6.3)	NA
Valproic acid	N03AG01	74186	45	0.9 (0.4)	7	NA	Yes	-1.4 (-9.0)	0.1 (-1.1)	2.2 (1.0)	1.0 (-4.0)	NA
Benzatropine	N04AC01	2666	7	2.5 (1.2)	2	NA	No	NA	1.6 (-1.6)	1.8 (-3.2)	NA	NA
Cabergoline	N04BC06	2918	5	1.9 (0.4)	2	2.7 (1.2)	No	NA	0.9 (-4.0)	1.1 (-6.5)	1.4 (-6.2)	NA
Lithium	N05AN01	25378	21	1.3 (0.6)	10	1.8 (0.8)	No	1.1 (-6.6)	0.7 (-1.2)	1.2 (-1.3)	1.3 (-3.7)	NA
Mefloquine	P01BC02	11290	9	1.2 (0.1)	6	2.4 (1.3)	No	NA	-0.7 (-4.6)	1.1 (-3.9)	NA	NA
Clenbuterol	R03CC63	299	3	2.6 (0.5)	2	2.7 (0.7)	No	NA	2.6 (-1.3)	NA	NA	NA

*Previous reports of association of drugs with myocarditis were looked for in the US Food and Drugs Administration labels as of January, 2020 (<https://labels.fda.gov/>).

Abbreviations: IC: information component, IC_{0.25}: lower bound of 95% credibility interval of the information component (significant when >0, in bold), IC_{0.005}: lower bound of 99.9% credibility interval of the information component (significant when >0, in bold), NA: No myocarditis case in this category; N_{observed}: number of myocarditis reports, N_{drug}: number of all reports related to the incriminated drug, N_{country}: number of countries from where myocarditis was reported; Tet: Tetanos; WHO ATC code: World Health Organization Anatomical Therapeutic Chemical code, v.: vaccine(s)

The authors also look to report the clinical features of all reported drug-induced myocarditis according to drug classes, which is an area where spontaneous reports can provide useful insights, as well as identifying potential new signals. More extensive evaluation should be conducted.

Answer: We have now added 2 tables providing the main clinical features and characteristics of suspected myocarditis by drug classes.

Added as Tables 2 and 3 (see after)

Table-2. Cases description, by drug substance

Substance	WHO ATC code	N substance	N observed	IC (IC ₀₂₅)	N country	Single suspected drug	N. fatalities	Women patients	Age in years	Delay in days	Previously reported*	First year with IC ₀₂₅ > 0
Antipsychotic												
Amisulpride	N05AL05	5917	14	2.6 (1.7)	4	0/14 (0.0%)	2/14 (14.3%)	7/14 (50.0%)	39 [25;41] ¹³	40 [7;74] ⁴	No	2015
Chlorpromazine	N05AA01	11224	12	1.6 (0.7)	4	1/12 (8.3%)	5/12 (41.7%)	4/12 (33.3%)	48 [32;53] ¹¹	40 [13;69] ⁶	Yes	1995
Clozapine	N05AH02	145208	3035	6.0 (5.9)	32	2843/3035 (93.7%)	153/3035 (5.0%)	698/2897 (24.1%)	36 [26;47] ²⁶²¹	17 [12;24] ¹¹⁹⁷	Yes	1993
Fluphenazine	N05AB02	4363	6	1.8 (0.4)	5	1/6 (16.7%)	5/6 (83.3%)	2/6 (33.3%)	33 [26;36] ⁶	52 [29;76] ²	No	2012
Haloperidol	N05AD01	30583	26	1.3 (0.7)	6	2/26 (7.7%)	5/26 (19.2%)	11/24 (45.8%)	41 [28;49] ²⁴	20 [12;31] ⁷	No	1995
Olanzapine	N05AH03	62298	60	1.5 (1.2)	13	11/60 (18.3%)	12/60 (20.0%)	21/58 (36.2%)	39 [29;48] ⁵¹	72 [12;296] ⁷	No	2013
Quetiapine	N05AH04	76947	67	1.4 (1.0)	12	19/67 (28.4%)	16/67 (23.9%)	24/60 (40.0%)	32.5 [23;41] ⁵⁶	27 [8;110] ⁹	Yes	2002
Zuclopenthixol	N05AF05	3803	7	2.1 (0.9)	3	1/7 (14.3%)	2/7 (28.6%)	4/7 (57.1%)	34 [31;45] ⁵	625 [313;936] ²	No	2018
Immunotherapy												
Aldesleukin	L03AC01	1575	16	4.0 (3.2)	2	14/16 (87.5%)	5/16 (31.3%)	2/13 (15.4%)	54 [44;60] ¹¹	na	Yes	2000
Atezolizumab	L01XC32	4762	33	4.0 (3.5)	9	21/33 (63.6%)	4/33 (12.1%)	14/31 (45.2%)	66.5 [60;74] ²⁴	33 [13;78] ¹⁰	Yes	2018
Avelumab	L01XC31	696	8	3.5 (2.4)	3	2/8 (25.0%)	2/8 (25.0%)	1/8 (12.5%)	68 [55;71] ⁸	52 [52;52] ⁴	Yes	2018
Cemiplimab	L01XC33	269	3	2.6 (0.5)	1	3/3 (100.0%)	0/3 (0.0%)	na	na	na	Yes	2019
Durvalumab	L01XC28	2611	16	3.6 (2.8)	7	11/16 (68.8%)	5/16 (31.3%)	2/14 (14.3%)	73.5 [68;76] ¹⁰	27 [20;30] ⁶	Yes	2018
Ipilimumab	L01XC11	18436	112	4.1 (3.8)	16	12/112 (10.7%)	50/112 (44.6%)	39/98 (39.8%)	69 [60;73] ⁷³	38 [21;54] ¹⁶	Yes	2015
Nivolumab	L01XC17	40718	300	4.5 (4.3)	22	186/300 (62.0%)	107/300 (35.7%)	100/277 (36.1%)	69 [61;75] ¹⁹⁸	34 [21;80] ⁷¹	Yes	2016
Pembrolizumab	L01XC18	21495	149	4.3 (4.1)	25	129/149 (86.6%)	48/149 (32.2%)	45/140 (32.1%)	70 [63;74] ¹⁰³	36 [18;105] ⁴⁹	Yes	2016
Cytotoxic												
Cyclophosphamide	L01AA01	93027	81	1.4 (1.1)	14	27/81 (33.3%)	32/81 (39.5%)	46/67 (68.7%)	43.5 [32;57] ⁶⁶	8 [5;22] ³¹	Yes	1999

Busulfan	L01AB01	5189	6	1.6 (0.2)	3	0/6 (0.0%)	5/6 (83.3%)	4/6 (66.7%)	42.5 [38;49] ⁴	12 [9;13] ⁴	Yes	2018
Cytarabine	L01BC01	29757	25	1.3 (0.7)	4	3/25 (12.0%)	8/25 (32.0%)	8/23 (34.8%)	47 [42;59] ²⁰	15 [11;26] ¹⁰	No	2015
Fluorouracil	L01BC02	73856	56	1.2 (0.8)	10	26/56 (46.4%)	3/56 (5.4%)	15/46 (32.6%)	51 [43;58] ⁴⁸	2 [2;15] ¹⁸	No	2009
Doxorubicin	L01DB01	74121	44	0.9 (0.4)	16	14/44 (31.8%)	15/44 (34.1%)	26/38 (68.4%)	54 [40;60] ³⁴	42 [11;120] ¹⁴	Yes	1985
Daunorubicin	L01DB02	7421	12	2.1 (1.2)	4	2/12 (16.7%)	1/12 (8.3%)	3/12 (25.0%)	40 [28;52] ¹¹	6 [5;27] ⁵	No	2014
Epirubicin	L01DB03	18370	12	1.0 (0.0)	3	1/12 (8.3%)	1/12 (8.3%)	4/4 (100.0%)	42.5 [39;48] ¹⁰	61 [61;61] ⁴	No	2010
Idarubicin	L01DB06	4004	22	3.6 (3.0)	4	3/22 (13.6%)	1/22 (4.5%)	8/19 (42.1%)	34.5 [24;47] ¹⁶	16 [12;23] ⁷	Yes	2010
Vaccine												
Anthrax vaccine	J07AC01	9588	163	5.5 (5.3)	1	7/163 (4.3%)	3/163 (1.8%)	2/163 (1.2%)	23 [21;28] ¹⁵⁵	11 [9;12] ¹⁴⁹	No	2010
Meningococcal vaccine	J07AH10	99170	54	0.7 (0.3)	12	15/54 (27.8%)	6/54 (11.1%)	8/53 (15.1%)	18 [16;20] ⁴⁸	5 [1;22] ²⁶	No	2012
Diphtheria vaccine;Pertussis vaccine; Tetanus vaccine	J07AJ52	204958	108	0.7 (0.4)	11	40/108 (37.0%)	32/108 (29.6%)	27/107 (25.2%)	15 [0;21] ⁹⁴	4 [2;10] ⁶¹	Yes	2014
Diphtheria vaccine;Tetanus vaccine	J07AM	46377	24	0.7 (0.0)	9	14/24 (58.3%)	2/24 (8.3%)	2/24 (8.3%)	30.5 [19;39] ²⁴	3 [1;6] ²⁰	Yes	2014
Typhoid vaccine	J07AP	14052	74	3.9 (3.5)	4	1/74 (1.4%)	2/74 (2.7%)	2/74 (2.7%)	22 [21;25] ⁷⁰	11 [8;12] ⁶⁷	No	2010
Tick-borne encephalitis vaccine	J07BA01	10449	13	1.8 (0.9)	3	11/13 (84.6%)	0/13 (0.0%)	3/13 (23.1%)	28 [21;43] ¹³	5 [3;21] ⁸	No	2011
Japanese encephalitis vaccine	J07BA02	4147	17	3.2 (2.5)	3	1/17 (5.9%)	1/17 (5.9%)	5/17 (29.4%)	20 [14;22] ¹⁵	10 [9;12] ¹³	No	2012
Influenza vaccine	J07BB02	231102	181	1.3 (1.0)	21	121/181 (66.9%)	28/181 (15.5%)	52/178 (29.2%)	38 [22;61] ¹⁶⁹	7 [2;13] ¹²²	No	1999
Hepatitis b vaccine	J07BC01	100233	71	1.1 (0.8)	11	34/71 (47.9%)	14/71 (19.7%)	26/70 (37.1%)	17.5 [1;34] ⁶⁴	7 [3;13] ⁴³	No	2010
Hepatitis a vaccine	J07BC02	42781	30	1.1 (0.5)	6	5/30 (16.7%)	4/30 (13.3%)	6/30 (20.0%)	19 [17;23] ²⁶	7 [3;11] ²⁰	No	2010
Hepatitis a vaccine;Hepatitis b vaccine	J07BC20	10195	13	1.8 (0.9)	5	13/13 (100.0%)	2/13 (15.4%)	2/13 (15.4%)	21 [21;37] ¹¹	16 [7;20] ⁸	No	2017
Smallpox vaccine	J07BX01	6767	383	7.2 (7.0)	4	195/383 (50.9%)	6/383 (1.6%)	12/383 (3.1%)	24 [21;28] ³⁶⁰	11 [9;13] ³⁴²	Yes	2010
Diphtheria vaccine;Polio vaccine;	J07CA01	6729	18	2.8 (2.0)	8	13/18 (72.2%)	0/18 (0.0%)	0/18 (0.0%)	23 [16;31] ¹⁸	3 [2;3] ¹²	Yes	2008

Tetanus vaccine													
Salicylate													
Aminosalicilic acid	A07EC	2501	5	2.1 (0.5)	3	5/5 (100.0%)	0/5 (0.0%)	1/4 (25.0%)	33 [23;35] ⁵	24 [21;27] ²	No	1997	
Sulfasalazine	A07EC01	24139	20	1.3 (0.6)	8	12/20 (60.0%)	3/20 (15.0%)	16/19 (84.2%)	34 [20;47] ¹⁹	17 [9;22] ⁵	Yes	1999	
Mesalazine	A07EC02	17129	311	5.7 (5.5)	26	281/311 (90.4%)	5/311 (1.6%)	77/300 (25.7%)	27.5 [20;39] ²⁶⁸	17 [12;28] ¹²⁷	Yes	1991	
Balsalazide	A07EC04	482	12	4.3 (3.3)	4	8/12 (66.7%)	1/12 (8.3%)	4/12 (33.3%)	28 [20;38] ¹¹	483 [121;2015] ⁴	No	2005	
Miscellaneous													
Stanozolol	A14AA02	442	3	2.4 (0.4)	2	1/3 (33.3%)	0/3 (0.0%)	2/3 (66.7%)	22 [22;23] ³	na	No	2017	
Norepinephrine	C01CA03	1863	4	2.0 (0.3)	2	4/4 (100.0%)	1/4 (25.0%)	3/4 (75.0%)	54.5 [28;82] ⁴	na	No	2018	
Dobutamine	C01CA07	2495	5	2.1 (0.5)	3	1/5 (20.0%)	0/5 (0.0%)	1/2 (50.0%)	49.5 [48;51] ²	na	No	2014	
Milrinone	C01CE02	978	5	2.7 (1.2)	2	0/5 (0.0%)	0/5 (0.0%)	0/1 (0.0%)	52 [52;52] ¹	na	No	2014	
Acitretin	D05BB02	3679	5	1.7 (0.2)	5	3/5 (60.0%)	0/5 (0.0%)	2/4 (50.0%)	37.5 [34;43] ⁴	61 [61;61] ²	No	2006	
Liothyronine	H03AA02	2484	5	2.1 (0.5)	3	1/5 (20.0%)	0/5 (0.0%)	2/4 (50.0%)	23.5 [22;25] ⁴	na	No	2017	
Minocycline	J01AA08	13162	25	2.4 (1.8)	7	24/25 (96.0%)	13/25 (52.0%)	18/24 (75.0%)	38 [18;46] ²³	19 [15;21] ⁶	Yes	2012	
Garenoxacin	J01MA19	969	3	2.1 (0.1)	1	1/3 (33.3%)	1/3 (33.3%)	2/3 (66.7%)	64 [53;69] ³	4 [4;4] ¹	No	2019	
Daptomycin	J01XX09	6462	8	1.7 (0.5)	2	2/8 (25.0%)	1/8 (12.5%)	1/7 (14.3%)	55 [49;55] ⁷	7 [6;7] ¹	No	2019	
Rituximab	L01XC02	77766	43	0.8 (0.3)	17	18/43 (41.9%)	13/43 (30.2%)	21/39 (53.8%)	57 [42;62] ³³	29 [11;147] ¹⁰	No	2009	
Trastuzumab	L01XC03	31989	27	1.3 (0.7)	12	5/27 (18.5%)	3/27 (11.1%)	15/16 (93.8%)	44.5 [39;53] ¹⁸	131 [76;267] ⁷	No	2010	
Arsenic trioxide	L01XX27	1857	4	2.0 (0.3)	3	0/4 (0.0%)	0/4 (0.0%)	2/4 (50.0%)	17 [17;17] ²	69 [43;96] ¹	No	2018	
Vemurafenib	L01XE15	8971	8	1.3 (0.1)	3	3/8 (37.5%)	0/8 (0.0%)	6/8 (75.0%)	48.5 [36;59] ⁸	26 [17;222] ⁴	No	2018	
Cobimetinib	L01XE38	1988	10	3.2 (2.2)	3	1/10 (10.0%)	0/10 (0.0%)	3/10 (30.0%)	60 [59;74] ⁹	17 [13;63] ⁷	Yes	2019	
Tretinoin	L01XX14	6044	20	3.1 (2.4)	6	7/20 (35.0%)	2/20 (10.0%)	10/17 (58.8%)	27 [19;36] ¹⁶	19 [16;21] ⁷	No	2014	
Valproic acid	N03AG01	74186	45	0.9 (0.4)	7	3/45 (6.7%)	9/45 (20.0%)	18/41 (43.9%)	44 [37;50] ³⁸	169 [72;271] ¹	Yes	2017	
Benzatropine	N04AC01	2666	7	2.5 (1.2)	2	0/7 (0.0%)	3/7 (42.9%)	4/6 (66.7%)	44 [33;44] ⁶	na	No	2016	
Cabergoline	N04BC06	2918	5	1.9 (0.4)	2	3/5 (60.0%)	0/5 (0.0%)	3/5 (60.0%)	46.5 [38;58] ⁴	818 [786;2096] ¹	No	2010	
Lithium	N05AN01	25378	21	1.3 (0.6)	10	7/21 (33.3%)	2/21 (9.5%)	7/18 (38.9%)	34 [28;47] ¹⁶	77 [19;1217] ⁵	No	2018	
Mefloquine	P01BC02	11290	9	1.2 (0.1)	6	8/9 (88.9%)	1/9 (11.1%)	1/9 (11.1%)	32 [30;57] ⁵	45 [29;56] ⁴	No	2019	
Clenbuterol	R03CC63	299	3	2.6 (0.5)	2	0/3 (0.0%)	0/3 (0.0%)	2/3 (66.7%)	22 [22;24] ³	na	No	2017	

*Previous reports of association of drugs with myocarditis were looked for in the US Food and Drugs Administration labels (<https://labels.fda.gov/>).

Continuous data are presented as median [interquartile range] ^{available data} and categorical data as number/available data (proportion).

Abbreviations: IC: information component, IC_{0.25}: lower bound of 95% credibility interval of the information component (significant when >0), N_{observed}: number of myocarditis reports, N_{drug}: number of all reports related to the incriminated drug, N_{country}: number of countries from where myocarditis was reported, N_{fatal}: number of declared deaths in myocarditis reports, na: not available, TTO: time to onset between first treatment intake and myocarditis, WHO ATC code: World Health Organization Anatomical Therapeutic Chemical code.

Table-3. Cases descriptions, by drug class, with heatmap of associated adverse drug reactions (green to red, least to most associated)

	Nobserved	Abdominal (aseptic, non-hepatic)	Anaphylaxis	Dermatologic	Endocrine	Eosinophilia	Heart failure	Hepato-biliary	Hydroelectrolytic imbalance	Hyperthermia	Infections	Kidney	Leucopenia	Myositis & myasthenia	Neurologic & psychiatric	Ophtalmologic	Osteoarticular & rheumatologic	Overdosage	Pericardial	Pseudo-coronary	Pulmonary (parenchymal)	Pulmonary (pleural)	Rhythmologic	Thrombocytopenia	Thromboembolism (peripheral)
Acitretin	5	0%	0%	60%	0%	20%	20%	40%	0%	40%	20%	20%	0%	20%	20%	0%	20%	0%	0%	20%	0%	20%	0%	0%	20%
Aldesleukin	16	31%	0%	13%	6%	0%	44%	13%	6%	6%	0%	19%	6%	13%	25%	0%	0%	0%	0%	38%	0%	0%	63%	6%	6%
Aminosalicilic acid	5	20%	0%	0%	0%	0%	40%	0%	0%	20%	0%	0%	0%	20%	0%	0%	0%	0%	20%	20%	40%	0%	0%	0%	0%
Amisulpride	14	7%	0%	14%	14%	0%	29%	14%	21%	14%	29%	0%	14%	14%	29%	0%	0%	14%	0%	50%	0%	0%	64%	0%	0%
Anthrax vaccine	163	3%	0%	5%	2%	1%	45%	4%	4%	18%	12%	2%	1%	23%	31%	1%	3%	0%	21%	90%	4%	1%	38%	1%	2%
Arsenic trioxide	4	0%	0%	0%	0%	0%	25%	0%	0%	0%	0%	0%	25%	0%	25%	0%	0%	0%	50%	25%	0%	0%	0%	0%	0%
Atezolizumab	33	3%	0%	3%	3%	0%	12%	3%	0%	0%	0%	0%	0%	12%	3%	6%	0%	0%	0%	9%	6%	0%	12%	6%	0%
Avelumab	8	0%	0%	0%	0%	0%	0%	0%	0%	0%	0%	0%	0%	25%	0%	0%	0%	0%	13%	13%	13%	0%	13%	0%	0%
Balsalazide	12	8%	0%	0%	0%	25%	8%	0%	0%	8%	0%	0%	0%	0%	8%	0%	0%	0%	33%	25%	8%	0%	0%	0%	0%
Benzatropine	7	0%	0%	14%	14%	14%	14%	0%	0%	43%	14%	14%	0%	0%	57%	0%	0%	43%	14%	29%	43%	14%	57%	0%	0%
Busulfan	6	17%	0%	17%	0%	0%	50%	17%	0%	0%	0%	33%	17%	0%	0%	0%	0%	0%	33%	17%	17%	0%	17%	0%	0%
Cabergoline	5	0%	0%	0%	40%	0%	60%	40%	0%	0%	0%	40%	0%	40%	40%	0%	40%	40%	20%	20%	20%	0%	40%	0%	40%
Cemiplimab	3	0%	0%	33%	0%	0%	0%	33%	0%	0%	33%	0%	0%	100%	0%	0%	0%	0%	0%	0%	33%	0%	0%	0%	0%
Chlorpromazine	12	0%	0%	8%	17%	8%	25%	8%	0%	33%	17%	8%	0%	8%	42%	0%	8%	17%	0%	33%	8%	8%	33%	0%	8%
Clenbuterol	3	33%	0%	0%	33%	0%	33%	33%	0%	33%	67%	0%	0%	33%	0%	0%	0%	67%	67%	33%	100%	0%	33%	0%	0%
Clozapine	3035	4%	0%	1%	2%	6%	16%	3%	1%	17%	8%	2%	2%	6%	13%	0%	1%	2%	7%	23%	5%	1%	26%	1%	1%
Cobimetinib	10	10%	0%	10%	0%	0%	10%	0%	0%	0%	0%	0%	0%	10%	0%	20%	0%	0%	0%	0%	0%	0%	0%	0%	0%
Cyclophosphamide	81	5%	4%	6%	4%	0%	44%	9%	1%	7%	10%	16%	7%	1%	7%	1%	5%	0%	15%	5%	9%	0%	11%	1%	2%
Cytarabine	25	4%	0%	0%	8%	0%	28%	8%	4%	8%	12%	16%	8%	4%	12%	0%	0%	4%	20%	20%	8%	4%	28%	8%	4%
Daptomycin	8	0%	0%	38%	0%	63%	25%	38%	13%	38%	50%	50%	0%	0%	0%	0%	0%	0%	13%	38%	13%	0%	50%	0%	0%
Daunorubicin	12	0%	0%	0%	8%	0%	8%	0%	0%	8%	17%	8%	0%	8%	25%	0%	0%	8%	8%	25%	17%	8%	17%	0%	0%

Diphtheria vaccine;Pertussis vaccine;Tetanus vaccine	108	4%	3%	7%	0%	2%	31%	6%	4%	40%	22%	4%	3%	19%	33%	1%	7%	1%	15%	41%	13%	3%	32%	0%	5%	
Diphtheria vaccine;Polio vaccine;Tetanus vaccine	18	0%	0%	0%	0%	6%	11%	6%	6%	56%	17%	0%	6%	22%	33%	0%	6%	6%	33%	28%	11%	11%	28%	6%	0%	
Diphtheria vaccine;Tetanus vaccine	24	4%	0%	13%	8%	0%	17%	4%	4%	54%	25%	8%	0%	17%	33%	4%	4%	0%	29%	50%	21%	0%	33%	4%	0%	
Dobutamine	5	0%	60%	0%	0%	0%	0%	0%	0%	0%	0%	0%	0%	0%	0%	0%	0%	0%	0%	0%	0%	0%	0%	0%	0%	0%
Doxorubicin	44	7%	7%	5%	0%	0%	45%	9%	5%	7%	18%	7%	7%	0%	11%	0%	0%	0%	11%	9%	7%	0%	14%	7%	2%	
Durvalumab	16	0%	0%	0%	0%	0%	6%	38%	0%	6%	0%	0%	0%	50%	0%	0%	6%	0%	0%	6%	25%	0%	6%	0%	6%	
Epirubicin	12	0%	0%	0%	0%	0%	83%	0%	0%	0%	0%	0%	0%	0%	0%	0%	0%	0%	0%	0%	0%	0%	0%	0%	0%	0%
Fluorouracil	56	2%	0%	0%	0%	0%	59%	0%	0%	0%	4%	0%	0%	0%	4%	0%	0%	0%	4%	29%	2%	0%	20%	0%	2%	
Fluphenazine	6	0%	0%	0%	33%	0%	17%	0%	0%	33%	17%	0%	0%	0%	50%	0%	0%	17%	0%	17%	17%	0%	50%	0%	17%	
Garenoxacin	3	0%	0%	0%	0%	0%	33%	33%	0%	0%	0%	0%	0%	0%	0%	0%	0%	0%	0%	33%	67%	0%	33%	0%	33%	
Haloperidol	26	4%	8%	4%	19%	0%	38%	12%	4%	23%	12%	8%	4%	15%	27%	4%	4%	12%	0%	15%	8%	0%	38%	0%	0%	
Hepatitis a vaccine	30	10%	0%	7%	7%	3%	13%	7%	3%	37%	7%	7%	7%	27%	33%	10%	27%	0%	20%	73%	7%	0%	27%	3%	0%	
Hepatitis a vaccine;Hepatitis b vaccine	13	23%	0%	8%	0%	0%	46%	31%	0%	46%	62%	15%	8%	8%	23%	0%	8%	0%	15%	38%	23%	0%	54%	8%	8%	
Hepatitis b vaccine	71	6%	0%	8%	6%	3%	34%	7%	4%	20%	25%	11%	0%	14%	34%	3%	4%	0%	14%	28%	13%	1%	34%	1%	4%	
Idarubicin	22	5%	0%	0%	14%	0%	36%	5%	0%	5%	23%	0%	9%	0%	5%	0%	0%	0%	14%	18%	5%	0%	27%	5%	5%	
Influenza vaccine	181	5%	0%	4%	3%	1%	36%	4%	4%	14%	25%	7%	1%	20%	30%	0%	3%	0%	14%	38%	15%	4%	35%	1%	2%	
Ipilimumab	112	6%	0%	5%	13%	0%	13%	17%	2%	4%	6%	12%	0%	28%	7%	2%	1%	2%	2%	10%	6%	3%	21%	2%	1%	
Japanese encephalitis vaccine	17	0%	0%	0%	0%	0%	24%	0%	0%	6%	0%	0%	0%	18%	29%	0%	0%	0%	35%	65%	0%	0%	29%	0%	0%	

Liothyronine	5	20%	0%	0%	40%	0%	40%	20%	0%	20%	40%	0%	0%	20%	0%	0%	0%	60%	40%	20%	60%	0%	20%	0%	0%
Lithium	21	10%	5%	5%	5%	10%	14%	10%	5%	14%	5%	10%	0%	14%	19%	5%	0%	24%	5%	14%	5%	0%	14%	0%	0%
Mefloquine	9	0%	11%	0%	0%	0%	22%	11%	0%	0%	11%	0%	0%	0%	22%	0%	0%	0%	11%	11%	0%	0%	11%	0%	0%
Meningococcal vaccine	54	2%	2%	2%	0%	2%	20%	2%	2%	43%	13%	0%	4%	22%	26%	2%	15%	0%	26%	56%	4%	0%	26%	0%	0%
Mesalazine	311	7%	2%	2%	1%	0%	12%	1%	1%	6%	3%	2%	0%	2%	5%	0%	1%	1%	14%	13%	3%	2%	7%	0%	2%
Milrinone	5	0%	20%	0%	0%	0%	0%	0%	0%	0%	0%	0%	0%	0%	0%	0%	0%	0%	0%	0%	0%	0%	0%	0%	0%
Minocycline	25	8%	12%	24%	20%	64%	40%	28%	0%	16%	36%	36%	0%	4%	12%	4%	8%	0%	12%	12%	12%	0%	28%	0%	16%
Nivolumab	300	4%	0%	4%	9%	0%	15%	14%	1%	2%	4%	6%	0%	31%	8%	2%	2%	0%	2%	8%	6%	2%	17%	1%	1%
Norepinephrine	4	0%	0%	0%	0%	0%	0%	0%	0%	0%	0%	0%	0%	0%	0%	0%	0%	0%	0%	0%	25%	0%	0%	0%	0%
Olanzapine	60	8%	2%	3%	7%	8%	33%	13%	5%	25%	22%	10%	2%	5%	47%	2%	2%	7%	8%	22%	8%	2%	38%	0%	5%
Pembrolizumab	149	3%	0%	3%	6%	0%	21%	12%	3%	1%	7%	3%	1%	29%	7%	1%	3%	1%	3%	7%	10%	2%	17%	1%	4%
Quetiapine	67	6%	3%	3%	4%	3%	19%	10%	0%	16%	7%	7%	4%	6%	34%	1%	1%	16%	3%	18%	6%	1%	33%	1%	0%
Rituximab	43	0%	5%	12%	12%	0%	26%	5%	2%	5%	28%	16%	5%	5%	5%	0%	12%	0%	2%	9%	9%	2%	12%	5%	7%
Smallpox vaccine	383	2%	0%	7%	2%	2%	39%	4%	3%	14%	11%	3%	0%	24%	25%	1%	2%	0%	25%	88%	4%	1%	43%	0%	1%
Stanozolol	3	33%	0%	0%	33%	0%	33%	33%	0%	33%	67%	0%	0%	33%	0%	0%	0%	33%	67%	33%	67%	0%	33%	0%	0%
Sulfasalazine	20	10%	5%	25%	5%	15%	10%	20%	0%	20%	5%	5%	0%	5%	0%	0%	5%	0%	25%	20%	5%	0%	10%	0%	0%
Tick-borne encephalitis vaccine	13	15%	0%	0%	0%	0%	0%	0%	0%	23%	23%	0%	0%	15%	23%	0%	8%	0%	8%	31%	0%	0%	15%	0%	0%
Trastuzumab	27	11%	7%	7%	0%	0%	74%	7%	0%	15%	11%	0%	11%	0%	15%	0%	0%	0%	4%	11%	7%	4%	19%	4%	4%
Tretinoin	20	0%	0%	0%	15%	0%	25%	0%	0%	0%	15%	0%	5%	0%	10%	0%	0%	0%	20%	15%	0%	0%	10%	0%	0%
Typhoid vaccine	74	1%	0%	4%	0%	3%	32%	4%	5%	16%	14%	5%	0%	22%	32%	0%	0%	0%	22%	84%	7%	1%	36%	1%	1%
Valproic acid	45	7%	0%	9%	11%	7%	22%	11%	4%	27%	16%	9%	9%	4%	29%	0%	7%	13%	13%	18%	22%	4%	36%	4%	0%
Vemurafenib	8	0%	0%	13%	0%	0%	38%	0%	0%	13%	0%	0%	0%	0%	0%	0%	0%	0%	0%	0%	13%	13%	13%	0%	0%
Zuclopenthixol	7	0%	0%	0%	0%	0%	14%	0%	0%	0%	0%	0%	0%	0%	57%	0%	0%	14%	0%	14%	0%	0%	14%	0%	0%

Re the TTO analysis, I think this is an important component of the manuscript. However the authors need to be careful in the assumptions they make around missing data being assumed to be the middle of the month and also missing data may well be systematically different to recorded data. With spontaneous reports one would expect suspected causality to be triggered if onset of adverse event is nearer to drug or vaccine exposure – and this likely bias should be stated explicitly and the anticipated likely impact on the results discussed.

Comparisons between the classes, e.g. in terms of age distributions can be interesting, but again caution is advised not to make too strong inference – differential chance and type of reporting (or missing data) could explain the differences that are seen, and the authors need to make the alternative explanations clear and motivate why they believe the differences represent genuine differences. TTO onset differences will be systematic between drug classes depending on the setting of drug prescribing/administration and the likelihood of careful follow up as reporting patterns are likely to vary by type and frequency of healthcare encounter. Similarly the probability of follow up information on mortality will vary greatly in a non-random way over time, between product, country etc.

Answer: Following the reviewer's recommendation, we added this potential bias and limitations in the Discussion section.

Added in Discussion: « *The fact that clinical presentations yielded from this work matched that of known drug-class-associated myocarditis is reassuring. However, TTO analyses, which also rely on spontaneous reporting, may suffer from reminiscing-bias, as a case more at chance to be related to a drug, if that drug was administered recently, as opposed to a drug administered years before. Hence, TTO may be underestimated, although comparison between observed TTO in our study and expected TTO from known literature did not yield significant differences. »*

Specific comments:

“A positive value of the IC025 is deemed significant”, is not statistically significant as in the same sense if it were a RCT or epidemiological study – I think important to write in full that means quantitative significantly more reporting than one would expect in terms of general reporting in the database ie it is relative to a background of other reports. The reason this is so important is that differential reporting patterns will affect the disproportionality (particularly for labelled events) and therefore quantitative scores need to be interpreted with caution. Potential issues such as masking are essential to consider. Disproportionality scores are predictive of signals, but clinical review is an essential follow up step.

Answer: We clarified this nuance, by transferring the disproportionality analyses details that we initially put in Supplementary Material, in the main manuscript and rewritten to emphasize the need to perform adequate clinical reviews, after a potential signal has been raised.

Added in Methods: *“Calculation of the IC using a Bayesian confidence propagation neural network was developed and validated by the Uppsala Monitoring Centre as a flexible, automated indicator value for disproportionate reporting that compares observed and expected drug–adverse drug reaction (ADR) associations to find new drug–ADR signals with identification of probability difference from the background data (full database).¹⁴ Probabilistic reasoning in intelligent systems (information theory) has proved to be effective*

for the management of large datasets, is robust in handling incomplete data, and can be used with complex variables. The information theory tool is ideal for finding drug-ADR combinations with other variables that are highly associated compared with the generality of the stored data.¹⁴ Several examples of validation with the IC exist, showing the power of the technique to find signals sooner after drug approval than by a regulatory agency, and to avoid false positives, whereby an association between a common drug and a common ADR occurs in the database only because the drug is widely used and the ADR is frequently reported (e.g., between atorvastatin and rash).^{14, 15} Like others, our team published several studies using VigiBase® and disproportional reporting calculation to characterize and identify new drug-ADR associated signals, which were subsequently corroborated by preclinical mechanistic studies or prospective cohorts.^{13, 16, 17, 18} This later element requires to be emphasized, as IC value should be interpreted only as means to perform clinical reviews of plausible associations and do not signify causality in any way. The $IC_{0.25}$ is the lower end of the 95% credibility interval for the IC. A positive value of the $IC_{0.25}$ is deemed significant. More information concerning calculation of the $IC/IC_{0.25}$ are provided in Supplementary-Material and have been recently used and detailed.^{13, 19}

“Drugs categorized as immunosuppressant (ATC label L04XX) were excluded to avoid indication bias.” Care is needed here of course – is there an risk of patients who had been co-administered LO4XX drugs had that information missing on the reports? Perhaps minimal chance, but useful to mention.

Answer: We have now added this limitation about information bias.

Added in Discussion: *“Moreover, not being able to return to each report to ensure that an exhaustive search for etiologies and concomitant drugs intake has been carried out leads to an information bias”*

The following statement is incorrect: “Of importance, two third of the drugs (38/62, 61.3%) reported in this analysis were not labeled as associated with myocarditis by the Food and Drug Administration and thus, represent new signals” they do not represent signals as used by WHO/FDA/EMA – they represent statistical alerts that require detailed clinical review to be considered safety signals. Disproportionality analyses do not “generate signals” they generate statistical alerts that require clinical review on a case level to be subsequently considered signals. The authors could use the term “signals of disproportional reporting” but would then have to explicitly mention the difference to “signals of suspected causality”.

Answer: We perfectly agree : this sentence was rephrased, to be toned down and better reflect the value of IC results.

Added to Discussion: *“Of importance, two third of the drugs (38/62, 61.3%) reported in this analysis were not labeled as associated with myocarditis by the Food and Drug Administration, and represent signals of suspected causality requiring further clinical reviews.»*

Similarly the authors conclude:” This study identified 62 drugs associated with myocarditis, 38 of which were not previously reported in FDA labels. They were grouped into: antineoplastic cytotoxics and immunotherapies, antipsychotics, salicylates, and vaccines. The time between treatment and myocarditis onset, presenting clinical features, subsequent mortality and patients’ profile varied significantly between

the 5 main drug classes.” However in a large database like Vigibase there are least some reports on virtually all possible drug/vaccine-AE reports: some with well-founded suspicion some only ill-founded. This conclusion is as currently phrased at best therefore not interesting and if ‘association’ is taken to allude to suspected causality might be misleading - recommend the authors restate to make their novel contribution more clear.

Answer: We agree, and rephrased the Conclusion to tone down association hypothesis as follow :

Added to Conclusion: « This study, based on disproportionality analyses, identified 62 drugs which may be associated with myocarditis. Among them, 38 were not previously reported in FDA labels. They were grouped into 5 categories: antineoplastic cytotoxics and immunotherapies, antipsychotics, salicylates, and vaccines. These categories presented distinct clinical presentation, time to onset and subsequent mortality, which suggest class effect. These elements warrant further clinical review to confirm association and causality. »

Recommendations

1. Subdivide the Myocarditis reports by drug into known and unknown.
 2. Focus all following analyses on the unknown subset, including the IC analysis
 3. Consider subgroup analysis if confounding/effect modification is suspected.
 4. Conduct detailed clinical review of potential emerging safety signals
 5. For emerging potential signal consider widening MEDDRA code selection to consider myocarditis cases erroneously captured with less specific codes
- Andrew Bate

Answer: We thank you for your time and hope that our revision fulfilled all requirements. The response to each specific point summarized in Recommendations has been addressed at the occasion of each specific question.

Respectfully,

Lee S. NGUYEN and Joe-Elie SALEM

REVIEWER COMMENTS

Reviewer #2 (Remarks to the Author):

I think the manuscript is interesting and feel that the manuscript is greatly improved in terms of wording around the IC/BCPNN and careful language about interpretation of disproportionality is now much more better and more clear throughout. Thank you.

I still feel however to be really as interesting and insightful to the reader as it should and could be more rationale of the clinical interpretation of the reports is needed as this is more important than the quantitative analyses.

It is clear that for these 62 drugs/vaccines that when a report is received in Vigibase for one of these drugs it is disproportionately more often a reporting of myocarditis (ie that these drugs have clearly positive IC values) than you'd expect based on general reporting patterns in the database. I do not yet feel yet convinced however though that all the 62 drugs/vaccines highlighted are signals of suspected causality of drug induced myocarditis. As the authors note and discuss there are a myriad of reporting patterns that might trigger the disproportionality and while discussing them is important I feel there now needs to be more emphasis and attention on what makes the authors believe that these are such signals ie that the particular nature of data on these reports are suggestive of drug-induced myocarditis, rather than alternative causes and/or explicable by confounding factors, much if which is not/cannot be accounted for in disproportionality scores. Explaining in more detail the clinical review of potential index individual cases for each drug/drug classes is critical to conclude signal is suspected causality. I appreciate there are too many cases for detailed discussion of all but summarising and then providing a more detailed discussion/assessment of the potential index cases so that it is clear how the descriptive data in the tables/supplementary information is considered from a causality perspective would be most valuable.

For each drug I would want to better understand: 1. Why you believe the myocarditis is likely to be drug-induced (not illfounded suspicion of a medicine due to e.g. temporality when actually it is virus), and 2. Why you believe this particular drug (if there are more than 1 on the report) is a likely culprit?

I would suggest some of the disproportionality discussion could now move to the supplement to give more focus to this clinical discussion (including temporality but also other aspects).

I think the extent to which plausible mechanisms can be described at least for drug groups, and is supported by case detail is an important element of the paper and should be added; even if some of the other material e.g. disproportionality work have to then be moved to the supplement. to reiterate, the strength of clinical suspicion that the reported details of case affords is what makes the paper, the disproportionality analysis is a weaker component.

Specific questions:

I find the more detailed review of the subset of French cases reassuring and strengthens the paper – could you just explain how you defined a PPV here?

You mention that despite limitations SRS is the best available data for post-marketing surveillance of myocarditis. Can you please add a sentence of two in the background to make clear why RWD is limited? This will make the reader more appreciate that while this analysis is limited it is an important contribution to the literature.

Some further introduction in the leadership between the distinction between drug-induced myocarditis and e.g. caused by virus – and differential diagnosis/presentation or not in terms of reporting would be very useful to contextualize your later results.

I didn't entirely follow the sensitivity analysis details in the methods – do you mean if two drugs are co-suspected and one has myocarditis labelled, that this would be excluded from the observed count for the other drug?

For the age subgroup test, I am less concerned about the risk of (and allowing for) multiple testing that non-random reporting into different age strata.

I know that some of the below are self-referential, but I do think that the articles looking at myocarditis in WHO database should be included and mentioned as its important for the reader how and over many years myocarditis has been studied in the WHO database, as this adds further credibility to this paper, please add at least one of:

Coulter, D.M., Bate, A., Meyboom, R.H., Lindquist, M. and Edwards, I.R., 2001. Antipsychotic drugs and heart muscle disorder in international pharmacovigilance: data mining study. *Bmj*, 322(7296), pp.1207-1209.

Hägg, S., Spigset, O., Bate A. and Söderström, T.G., 2001. Myocarditis related to clozapine treatment. *Journal of clinical psychopharmacology*, 21(4), pp.382-388.

Noseda, R., Ruinelli, L., Gaag, L.C. and Ceschi, A., 2020. Pre-Existing Cardiovascular Conditions as Clinical Predictors of Myocarditis Reporting with Immune Checkpoint Inhibitors: A VigiBase Study. *Cancers*, 12(11), p.3480.

i hope you find this helpful, Best wishes Andrew

Answer to reviewers – R2

Dear Reviewers

We are honored that our manuscript retained your attention for this second round of revisions.

We performed and addressed all queries asked by the reviewer and feel this last version of the manuscript is more fluid and relevant to the fellow reader of your esteemed journal, Nature Communications. All modifications and specific answers to the reviewer are included in the following pages, in blue color.

We hope that you will find it suitable for publication and thank you again for your time and benevolence.

Kindly,

Lee S. NGUYEN, MD, PhD

&

Joe-Elie SALEM, MD, PhD

I think the manuscript is interesting and feel that the manuscript is greatly improved in terms of wording around the IC/BCPNN and careful language about interpretation of disproportionality is now much more better and more clear throughout. Thank you.

I still feel however to be really as interesting and insightful to the reader as it should and could be more rationale of the clinical interpretation of the reports is needed as this is more important than the quantitative analyses.

It is clear that for these 62 drugs/vaccines that when a report is received in Vigibase for one of these drugs it is disproportionately more often a reporting of myocarditis (ie that these drugs have clearly positive IC values) than you'd expect based on general reporting patterns in the database. I do not yet feel yet convinced however though that all the 62 drugs/vaccines highlighted are signals of suspected causality of drug induced myocarditis. As the authors note and discuss there are a myriad of reporting patterns that might trigger the disproportionality and while discussing them is important I feel there now needs to be more emphasis and attention on what makes the authors believe that these are such signals ie that the particular nature of data on these reports are suggestive of drug-induced myocarditis, rather than alternative causes and/or explicable by confounding factors, much of which is not/cannot be accounted for in disproportionality scores. Explaining in more detail the clinical review of potential index individual cases for each drug/drug classes is critical to conclude signal is suspected causality. I appreciate there are too many cases for detailed discussion of all but summarizing and then providing a more detailed discussion/assessment of the potential index cases so that it is clear how the descriptive data in the tables/supplementary information is considered from a causality perspective would be most valuable.

For each drug I would want to better understand: 1. Why you believe the myocarditis is likely to be drug-induced (not ill-founded suspicion of a medicine due to e.g. temporality when actually it is virus), and 2. Why you believe this particular drug (if there are more than 1 on the report) is a likely culprit?

Following your suggestions, we enhanced association hypotheses by performing additional pharmacovigilance analyses. Notably, in drugs which were not previously known as associated with myocarditis (as per FDA labels), we assessed magnitude of causality, with the corresponding score. Using standardized pharmacovigilance causality assessment scoring, we attributed a Chronological score, Semiological score, which, combined yield the Intrinsic imputability score; and independently, we provided a Bibliography score (i.e. extrinsic accountability). We added the method in the relevant section:

“Additionally, we performed a pharmacovigilance causality assessment analysis following the French method, on all drugs which were not previously described associated with myocarditis, nor shared similar pharmacological properties as drugs which were known to be associated with myocarditis. This analysis was based on three criteria: chronological, semiological and extrinsic accountability. Chronological criterion score corresponds to: C0: incompatible, C1: doubtful, C2: plausible, C3: probable. Semiological criterion score, which is based on semiotics, drug dechallenge or rechallenge, and existence of confounding elements (preexisting co-morbidities or co-medications) corresponds to: S1: questionable, S2: plausible, S3: likely. Imputability score combines chronological and semiological criteria and corresponds to: I0: incompatible, I1: doubtful, I2: plausible, I3: likely and I4: very likely. Finally, extrinsic accountability is a bibliography score: B0: unpublished, B1: class effect, B2: widely published and B3: expected effect (described in the product information).¹⁷”

Supplementary Table D. Pharmacovigilance causality assessment of drug substances not previously associated with myocarditis in their FDA label description

Substance	WHO ATC code	Nsubst	Nobserved	IC (IC025)	Single suspected drug	Nfatal	Age in years	Delay in days	Previously reported	Chronological criterion	Semilogic criterion	Intrinsic imputability	Extrinsic accountability (Bibliography score)
Stanozolol	A14AA02	442	3	2,4 (0,4)	1/3 (33,3%)	0/3 (0,0%)	22 [22;23] 3		No	NA	2	NA	2
Norepinephrine	C01CA03	1863	4	2,0 (0,3)	4/4 (100,0%)	1/4 (25,0%)	54,5 [28;82] 4		No	NA	1	NA	1
Dobutamine	C01CA07	2495	5	2,1 (0,5)	1/5 (20,0%)	0/5 (0,0%)	49,5 [48;51] 2		No	NA	1	NA	0
Milrinone	C01CE02	978	5	2,7 (1,2)	0/5 (0,0%)	0/5 (0,0%)	52 [52;52] 1		No	NA	1	NA	0
Acitretin	D05BB02	3679	5	1,7 (0,2)	3/5 (60,0%)	0/5 (0,0%)	37,5 [34;43] 4	61 [61;61] 2	No	1	2	1	2
Liothyronine	H03AA02	2484	5	2,1 (0,5)	1/5 (20,0%)	0/5 (0,0%)	23,5 [22;25] 4		No	NA	2	NA	2
Garenoxacin	J01MA19	969	3	2,1 (0,1)	1/3 (33,3%)	1/3 (33,3%)	64 [53;69] 3	4 [4;4] 1	No	3	2	3	2
Daptomycin	J01XX09	6462	8	1,7 (0,5)	2/8 (25,0%)	1/8 (12,5%)	55 [49;55] 7	7 [6;7] 1	No	3	2	2	2
Rituximab	L01XC02	77766	43	0,8 (0,3)	18/43 (41,9%)	13/43 (30,2%)	57 [42;62] 33	29 [11;147] 10	No	2	2	2	2
Trastuzumab	L01XC03	31989	27	1,3 (0,7)	5/27 (18,5%)	3/27 (11,1%)	44,5 [39;53] 18	131 [76;267] 7	No	1	1	1	2
Arsenic trioxide	L01XX27	1857	4	2,0 (0,3)	0/4 (0,0%)	0/4 (0,0%)	17 [17;17] 2	69 [43;96] 1	No	1	1	1	2
Vemurafenib	L01XE15	8971	8	1,3 (0,1)	3/8 (37,5%)	0/8 (0,0%)	48,5 [36;59] 8	26 [17;222] 4	No	3	2	2	2
Tretinoin	L01XX14	6044	20	3,1 (2,4)	7/20 (35,0%)	2/20 (10,0%)	27 [19;36] 16	19 [16;21] 7	No	3	2	3	3
Benzatropine	N04AC01	2666	7	2,5 (1,2)	0/7 (0,0%)	3/7 (42,9%)	44 [33;44] 6		No	NA	3	NA	2
Cabergoline	N04BC06	2918	5	1,9 (0,4)	3/5 (60,0%)	0/5 (0,0%)	46,5 [38;58] 4	818 [786;2096] 1	No	1	3	1	2
Lithium	N05AN01	25378	21	1,3 (0,6)	7/21 (33,3%)	2/21 (9,5%)	34 [28;47] 16	77 [19;1217] 5	No	2	2	2	2
Mefloquine	P01BC02	11290	9	1,2 (0,1)	8/9 (88,9%)	1/9 (11,1%)	32 [30;57] 5	45 [29;56] 4	No	2	2	2	2
Clenbuterol	R03CC63	299	3	2,6 (0,5)	0/3 (0,0%)	0/3 (0,0%)	22 [22;24] 3		No	NA	2	NA	2

Chronological criterion score corresponds to: C0: incompatible, C1: doubtful, C2: plausible, C3: probable; Semiological criterion corresponds to: S1: questionable, S2: plausible, S3: likely; Imputability score corresponds to: I0: incompatible, I1: doubtful, I2: plausible, I3: likely and I4: very likely and Bibliography score corresponds to: B0: unpublished, B1: class effect, B2: widely published and B3: expected effect (described in the product information)

I would suggest some of the disproportionality discussion could now move to the supplement to give more focus to this clinical discussion (including temporality but also other aspects).

We thank the reviewer for this suggestion, albeit after rephrasing all elements in the Discussion and Methods section, we managed to remain within the words count limit. Moreover, we feel that previous queries asked by this reviewer regarding Disproportionality, which we added in the previous revision, are still relevant in the main manuscript.

I think the extent to which plausible mechanisms can be described at least for drug groups, and is supported by case detail is an important element of the paper and should be added; even if some of the other material e.g. disproportionality work have to then be moved to the supplement. to reiterate, the strength of clinical suspicion that the reported details of case affords is what makes the paper, the disproportionality analysis is a weaker component.

As suggested, we expanded in the Discussion section regarding the mechanisms which have been hypothesized as causes of myocarditis, in all major drug classes that we described, as follow (references are self-contained in the main manuscript):

“Antipsychotic antidopaminergic drugs such as clozapine are probably linked to a type-1-immunoglobulin-E-mediated hypersensitivity reaction, and anticholinergic blockade with high sympathetic drive responsive to beta-adrenergic blockade. There have been several proposed mechanisms for salicylate-associated myocarditis: direct toxicity on the myocardium, allergic reaction mediated by immunoglobulin E, cell-mediated hypersensitivity reaction, or a humoral antibody response. Both antipsychotic agents and salicylates are associated with eosinophilic myocarditis, and were the two classes of drugs most frequently associated with eosinophilia in this work. Immunotherapies have been associated with fulminant lymphocytic myocarditis, due to immune-checkpoint inhibition that is specifically mediated by T cells. Preclinical models with PD1 and CTLA-4 gene deletion manifest severe myocarditis, while histology in human heart presenting with immunotherapy-induced myocarditis show T-cells and macrophages infiltrates resembling cardiac allograft cellular rejection. Cytotoxic-agents used as antineoplastic also feature direct cytotoxicity to cardiomyocytes with myofibrillar disarray due to neuregulin 1β dysregulation, associated with mitochondrial apoptosis and free radical production mechanisms. Finally, vaccine (most prominently smallpox) are associated with myocarditis, mediated by autoimmunity secondary to vaccine-mimicry of myocardium antigens and more recently, activation of toll-like receptors have been more specifically discussed.”

Specific questions:

I find the more detailed review of the subset of French cases reassuring and strengthens the paper – could you just explain how you defined a PPV here?

As included in the first revision, we provided the definition of PPV in the Methods section: “Cases were retrospectively analyzed to compute the **positive predictive value (true positive/(true positive+false positive))** of clinically-suspected myocarditis, as defined by the ESC guidelines”

You mention that despite limitations SRS is the best available data for post-marketing surveillance of myocarditis. Can you please add a sentence of two in the background to make clear why RWD is limited? This will make the reader more appreciate that while this analysis is limited it is an important contribution to the literature.

We thank the reviewer for his suggestion. We added the following sentence in the Methods:
“VigiBase® is a spontaneous reporting system which allows for more robust and rigorous analyses than isolated case reports or case series, due to the possibility of performing quantitative comparisons, such as disproportionality analysis (case–non-case) to identify drugs significantly associated with myocarditis.”

Some further introduction in the leadership between the distinction between drug-induced myocarditis and e.g. caused by virus – and differential diagnosis/presentation or not in terms of reporting would be very useful to contextualize your later results.

As suggested, we provided an additional table regarding the plausibility of causality between suspected drugs and myocarditis, when the incriminated drug had not been previously flagged associated with myocarditis in its FDA label. It has to be noted that the Semiologic component score includes the fact that patient presented with a disease likely (or not) to be associated with myocarditis.

I didn't entirely follow the sensitivity analysis details in the methods – do you mean if two drugs are co-suspected and one has myocarditis labelled, that this would be excluded from the observed count for the other drug?

Indeed, as an additional sensitivity analysis in the revised manuscript (Supp Table C), we performed the same disproportionality analyses in a subset in which we systematically excluded cases which contained already known drugs associated with myocarditis, to avoid emergence of drug signals triggered by biased co-prescription with a known liable drug (e.g concomitant use of antipsychotic drugs with clozapine).

For the age subgroup test, I am less concerned about the risk of (and allowing for) multiple testing that non-random reporting into different age strata.

Indeed, we used the IC_{0005} as a threshold, as recommended by the Uppsala Monitoring Center (UMC), to ascertain significant associations, in these subgroup analyses (<https://www.who-umc.org/vigibase/vigilyze/analytics-in-vigilyze/>).

I know that some of the below are self-referential, but I do think that the articles looking at myocarditis in WHO database should be included and mentioned as its important for the reader how and over many years myocarditis has been studied in the WHO database, as this adds further credibility to this paper, please add at least one of:

Coulter, D.M., Bate, A., Meyboom, R.H., Lindquist, M. and Edwards, I.R., 2001. Antipsychotic drugs and heart muscle disorder in international pharmacovigilance: data mining study. *Bmj*, 322(7296), pp.1207-1209.

Hägg, S., Spigset, O., Bate A. and Söderström, T.G., 2001. Myocarditis related to clozapine treatment. *Journal of clinical psychopharmacology*, 21(4), pp.382-388.

Noseda, R., Ruinelli, L., Gaag, L.C. and Ceschi, A., 2020. Pre-Existing Cardiovascular Conditions as Clinical Predictors of Myocarditis Reporting with Immune Checkpoint Inhibitors: A VigiBase Study. *Cancers*, 12(11), p.3480

I hope you find this helpful,

Best wishes Andrew

Following your suggestions, we added all three papers and modified the Introduction to include them:

“In contrast, drug toxicity and hypersensitivity are underdiagnosed causes of myocarditis, that may be responsible for severe and complex clinical presentation, including fulminant lymphocytic myocarditis,^{1, 5, 6} and allergic or hypersensitivity eosinophilic myocarditis.⁷ While imperfect, pharmacovigilance analyses based on real world evidence coming from spontaneous report systems allow for post-marketing drug surveillance (i.e. phase IV), and historically identified classes of drugs associated with myocarditis: immune checkpoint inhibitors, antipsychotics, antibiotics, and vaccines.”

On a side note, we commend these works which inspired us to write this paper. However, we must emphasize that the degree of causality was not as thoroughly explored in these papers as in the one we would like to publish, and we thank once again Mr Bates to have pushed us to go deeper into this analysis.

Regarding the reference of Nosedá et al (Cancers, 12(11), p.3480), we must also stress out that Nosedá et al. performed a follow-up of our own work. At the time, we did not deem it relevant to analyze pre-existing cardiovascular diseases, as this element was not adequately described in VigiBase[®], all the more so that cardiovascular treatments are not very informative towards the kind of cardiovascular disease it may treat (for example, betablockers may be indicated in both hypertension and heart failure with reduced ejection fraction, precluding us from concluding).

To conclude, we thank Mr Bates for his insights and benevolence. We feel that the paper written as is, in its third iteration, after 7 months of reviewing, has never been so clear. Should you require additional analyses, we would be more than thrilled to collaborate in any future work.

Respectfully,

Lee S. NGUYEN and Joe-Elie SALEM

REVIEWERS' COMMENTS

Reviewer #2 (Remarks to the Author):

I agree w the authors - this manuscript has greatly improved in depth and insights. I thank them for all their work to improve it which has certainly paid dividends.

Some comments on the revised manuscript:

I see too little discussion of the causality assessment that the authors have conducted throughout (abstract, discussion and elsewhere). The pairs identified with the IC that had strong causality assessments are the signals - please don't refer to the 62 as 'signals' they are merely statistical alerts. The combination of disproportionality with stronger causality assessment scores for unlabelled is where the interest would lie.

Please make clear in the manuscript that pre-existing cardiovascular diseases are not well captured in this data source and this is an insurmountable limitation of this analysis, therefore. Although while the reader should be aware of this limitation the results are still very interesting.

My question about why SRS data was used was not adequately addressed: Instead of an additional sentence on the use of SRS, I think we need a clear sentence to address the earlier comment that there is no better source than SRS to address this topic. So for example why was this study not better conducted in RWD such as SNIIRAM - as this will be an obvious question for the reader.

Answer to the reviewers

Reviewer 2

I agree w the authors - this manuscript has greatly improved in depth and insights. I thank them for all their work to improve it which has certainly paid dividends.

Some comments on the revised manuscript:

I see too little discussion of the causality assessment that the authors have conducted throughout (abstract, discussion and elsewhere). The pairs identified with the IC that had strong causality assessments are the signals - please don't refer to the 62 as 'signals' they are merely statistical alerts. The combination of disproportionality with stronger causality assessment scores for unlabelled is where the interest would lie.

Indeed, we toned down our results and removed the word 'signal' referring to those 62.

Please make clear in the manuscript that pre-existing cardiovascular diseases are not well captured in this data source and this is an insurmountable limitation of this analysis, therefore. Although while the reader should be aware of this limitation the results are still very interesting.

Limitations were expanded to include lack of pre-existing comorbidities, including cardiovascular diseases: *"Moreover, pre-existing cardiovascular diseases are not exhaustively collected in this data source, as only drugs and their indications are mentioned, while existing comorbidities which may not be treated cannot be reported."*

My question about why SRS data was used was not adequately addressed: Instead of an additional sentence on the use of SRS, I think we need a clear sentence to address the earlier comment that there is no better source than SRS to address this topic. So for example why was this study not better conducted in RWD such as SNIIRAM - as this will be an obvious question for the reader.

The topic about SRS versus RWD data has now been addressed with the following paragraph: *"Global pharmacovigilance systems rely on spontaneous reporting systems, which provide a large volume of information and allow for the early detection of issues related to drugs or their use. While not without flaws, these systems are specifically designed to capture the information related to adverse drug reaction with dedicated and focused data collection concerning the treatment modalities. On the other hand, real world data coming from administrative database used for reimbursement of care, such as the French Système national d'information inter-régimes de l'Assurance maladie (SNIIRAM), may have larger volumes of data. However, in the latter, quality of data is driven by economic and administrative focus with lack of basic information (duration, effective start and end date of drug intake) and lack of information of drugs which are not reimbursed."*